# Distinct S-adenosylmethionine synthases link phosphatidylcholine to mitochondrial function and stress survival

Athena L. Munden[1], Arjamand Mushtaq[1], Dominique S. Lui[1], Katherine M. Edwards[1], Daniel P. Higgins[1], Kasturi Biswas[1], Rachel M. Walker[1], Leah H. Crowley[1], Matthew J. Fanelli[1], Thien-Kim Ngyuen[1], Maria Ericsson[2], Adwait A. Godbole[1], John A. Haley[1], Caroline A. Lewis[1], Jessica B. Spinelli[1,3], Benjamin Harrison[4,5], Daniel Raftery[6], Danijel Djukovic[6], Daniel E. L. Promislow[4,5¤], Dana L. Miller[7☉], Amy K. Walker [1]*☉

1 Program in Molecular Medicine, UMASS Chan Medical School, Worcester, Massachusetts, United States of America, 2 Harvard Medical School, Boston, Massachusetts, United States of America, 3 Howard Huges Medical Institute Freeman Hrabowski Scholar, University of Massachusetts Chan Medical School, Worcester, Massachusetts, United States of America, 4 Department of Laboratory Medicine and Pathology, University of Washington, School of Medicine, Seattle, Washington, United States of America, 5 Department of Biology, University of Washington, School of Medicine, Seattle, Washington, United States of America, 6 Department of Anesthesiology and Pain Medicine, University of Washington Medicine School of Medicine, Seattle, Seattle, Washington, United States of America, 7 Department of Biochemistry, University of Washington School of Medicine, Seattle, Washington, United States of America

☉ These authors contributed equally to this work.
¤ Current address: Jean Mayer USDA Human Nutrition Research Center on Aging, Tufts University, Boston, Massachusetts, United States of America
* amy.walker@umassmed.edu

## Abstract

S-adenosylmethionine (SAM), produced by SAM synthases, is critical for various cellular regulatory pathways and the synthesis of diverse metabolites. Humans and many other organisms express multiple SAM synthases. However, loss of different synthase activity can have distinct phenotypic effects. For instance, in *Caenorhabditis elegans* loss of *sams-1* leads to enhanced heat shock survival and increased life span, but loss of *sams-4* reduces heat stress survival. This provides a biological context to test the hypothesis that the enzymatic source of SAM impacts its function and to identify mechanistic connections. Here, we show that SAMS-1 contributes SAM to a variety of intermediary metabolic pathways, whereas SAMS-4 has a more limited role to support SAM-dependent protein transmethylation reactions. Mitochondria seem to be particularly impacted specifically by loss of *sams-1*; many mitochondrial metabolites are perturbed and there is an age-dependent decline of nuclear-encoded mitochondrial gene expression in these animals. We further demonstrate that reduced production of phosphatidylcholine in *sams-1*-deficient animals leads to mitochondrial fragmentation and subsequent loss of mitochondrial components. We propose that alterations in mitochondria are mechanistically linked to the increased survival in heat stress specific to *sams-1*-deficient animals.

**Data availability statement:** All relevant data are in the paper except for the RNA sequencing data, which is accessible at the GEO under the accession numbers GSE288260 and GSE308504. Archived code is located on Zenodo (https://doi.org/10.5281/zenodo.17495841).

**Funding:** This study was supported by the National Institutes of Health grant R01AG0686701 (AKW), the National Institutes of Health grant 1R01AG053355 (AKW, DLM), the United States Department of Agriculture cooperative agreement USDA/ARS 58-8050-9-004 (DP), the National Institutes of Health grant University of Washington Nathan Shock Center of Excellence in the Basic Biology of Aging (NIH P30AG013280) (DP), the National Institutes of Health grant 1R01DK144534 (JBS), the Searle Scholar's Program (JBS), the AFAR Junior Faculty Award (JBS), and the Smith Family Foundation (JBS). The funders had no role in study design, data collection and analysis, decision to publish, or preparation of the manuscript.

**Competing interests:** The authors have declared that no competing interests exist.

**Abbreviations:** 1CC, 1-carbon cycle; CL, cardiolipin; CV, coefficient of variation; FS, Freeze Substitution; LC–MS, liquid chromatography–mass spectrometry; OCR, oxygen consumption rate; PC, phosphatidylcholine; PCA, principal component analysis; PGs, phosphatidylglycerols; PO, propylene oxide; QC, quality control; SAM, S-adenosylmethionine; TCA, tricarboxylic acid cycle; TEM, transmission electron microscopy; TMRE, tetramethylrhodamine, ethyl ester; UPR, unfolded protein response.

## Introduction

Remodeling gene expression to adapt to distinct physiological conditions is a common strategy for countering exogenous stressors [1]. The transcriptional response to stress depends on the nature of the stressor; defensive peptides are produced in response to pathogen stress, detoxification enzymes are upregulated in oxidative stress, and chaperones are expressed upon protein folding stress. The metabolic state of the cell also influences the transcriptional response to stress [2]. For example, limitation of the 1-carbon cycle (1CC) metabolite S-adenosylmethionine (SAM) has distinct effects on stress responses, depending on the type of stress and the metabolic source of SAM [3–5]. SAM could influence gene expression through its role as the predominant cellular methyl donor, by supporting histone methylation and chromatin regulatory patterns. However, SAM also contributes to other metabolic pathways, including those that generate phosphatidylcholine (PC), glutathione, and polyamines. Perturbations of these pathways may also have indirect effects on gene expression [6,7]. This pleiotropy underscores the need to identify mechanistic links between SAM, its downstream metabolites or methylation targets, and phenotypes resulting from SAM depletion, such as altered stress responses, lipid accumulation, changes in differentiation or development, and extended life span [3,8–13].

SAM is produced from methionine and ATP by SAM synthase, a highly conserved enzyme present as multiple paralogs in eukaryotes. Yeast and mammals both have two synthases (SAM1/SAM2 and MAT1A/MAT2A, respectively). In yeast, the synthases have distinct effects on gene expression and genome stability [14]. In mammals, SAM synthases are expressed in different tissues. MAT1A is specific to adult liver and KO mice have fatty liver while MAT2A is expressed widely [15]. MAT2A also has a regulatory partner, MAT2B, which can form multiple heteromeric complexes with distinct enzymatic characteristics [15]. However, understanding the mechanistic properties of these synthases or isoforms has been difficult to study in mammals, as MAT1A expression decreases in cultured cells, MAT2A is essential for viability, and cell culture media is replete with 1CC intermediates [16].

We have focused on SAM synthases in *Caenorhabditis elegans*, where the family has been extended to 4 paralogs, *sams-1 (X)*, the highly similar *sams-3* and *sams-4 (IV)* genes that are expressed bicistronicly from the same promoter, and *sams-5 (IV)*. Most somatic cells express at least two *sams* genes [5]. Knockdown of each distinct synthase has similar effects on total SAM levels [3,5,8], but induces different phenotypes. This provides a robust biological context for exploring mechanisms linking specific SAM synthases to phenotypes. Animals lacking *sams-1* have increased lipid stores [8], an extended life span [9], and have complex stress phenotypes. Loss of *sams-1* sensitizes animals to bacterial stress [3], but these animals live longer after heat shock [4,5]. In contrast, reduction in *sams-4* increases sensitivity to heat shock, but has negligible effects on life span [5] or lipogenic gene expression [5]. Global H3K4me3 patterns change after RNAi of either *sams-1* or *sams-4,* but only *sams-4* is sensitive to heat stress, mimicking the poor survival after RNAi knockdown of the H3K4 methyltransferase *set-16*/MLL [4,5]. In heat shock, *sams-4* is required for

increased production of SAM and H3K4me3 in the absence of *sams-1* [5], further supporting a scenario where production of SAM from SAMS-4 ($SAM_4$) is associated with stress-induced changes in chromatin methylation patterns important for heat shock survival.

In this study, we took advantage of these phenotypic distinctions to identify the metabolic products and cellular mechanisms that connect SAM-synthase specific production to the biology of stress and aging. Here, we show that animals with reduced SAM synthesized by SAMS-1 ($SAM_1$), remodel mitochondrial metabolism, morphology, and function. We link $SAM_1$ specifically to the production of PC, a membrane lipid impacting curvature and organelle function. We find that $SAM_1$ acts through PC production to influence mitochondrial fission, entry into the mitophagy pathway and influence stress survival. Collectively, our results support a model where different SAMS enzymes provision specific methylation reactions or metabolic pathways that have independent physiological effects.

## Results

### SAMS-1, but not SAMS-4, affects a variety of cellular metabolites

The *sams-1* and *sams-4* genes both encode SAM synthases (Fig 1A, see Fig 2A for a Methionine/SAM cycle diagram). Depletion of *sams-1* and *sams-4* results in different phenotypes, suggesting that SAM produced by each enzyme may function in distinct cellular processes or metabolic pathways. To explore this possibility, we performed targeted metabolomics in animals where SAM production was reduced by knockdown of either *sams-1* or *sams-4*. We identified 62 metabolites that significantly differed across conditions (S1 Table). S1 Fig shows the relationships between selected metabolites using pathway diagrams based on WormPaths [17]. Metabolites will be referred to with abbreviations as defined in WormPaths throughout this manuscript.

In unstressed animals, principal component analysis (PCA) shows widespread changes in metabolites in the *sams-1(lof)* animals, whereas the *sams-4(lof)* metabolite profile was quite similar to controls (Fig 1B, 15°C). These changes mirror the effects of *sams-1* and *sams-4* on gene expression [5]: knockdown of *sams-1* by RNAi had widespread effects, whereas *sams-4(RNAi)* animals were more similar to controls. We next analyzed the metabolite profiles of animals exposed to heat stress, where *sams-4(lof)* die rapidly but *sams-1(lof)* survive better than wild-type [5]. We identified 46 metabolites that significantly changed in animals after heat stress. Inspection of PCA plots indicate there were dramatic changes in metabolites upon heat stress in wild-type animals. The metabolite profile of *sams-4(RNAi)* animals after heat stress matched the changes in wild-type, whereas we observed broad and distinct changes in *sams-1(RNAi)* animals (Fig 1B, 37°C). The limited effects of *sams-4(RNAi)* on metabolite abundances is consistent with the idea that its role in histone methylation is more salient to the observed decreased survival of *sams-4(lof)* animals in heat stress. Further supporting this idea, the changes in gene expression we observe in *sams-4(RNAi)* animals exposed to heat stress [5] mirror changes observed when the histone methyltransferase *set-16* is depleted [4], suggesting that these changes in gene expression are related to histone methylation status. In contrast, our data indicate that increased survival of *sams-1(lof)* animals in heat stress is associated with both metabolic changes in addition to altered gene expression.

To define the specific effects of *sams-1* and *sams-4*, we identified the metabolites that changed when each SAMS enzyme was depleted. Consistent with our PCA, we observed many changes in metabolite abundance in *sams-1(RNAi)* animals in basal conditions (Fig 1C) but only a few metabolites decreased in *sams-4(RNAi)* animals (Fig 1D). Importantly, we found that steady-state SAM (amet) abundance decreased by ~50% after RNAi of either *sams-1* or *sams-4* (Fig 2B), consistent with previous results [3,5,8]. We conclude that the differences we observe between *sams-1* and *sams-4* animals do not result from quantitative changes in SAM production, but rather from the downstream effects of the loss of a specific source of SAM. Moreover, the fact that depletion of either SAMS enzyme reduces the abundance of SAM indicates that these enzymes do not compensate for each other in basal conditions.

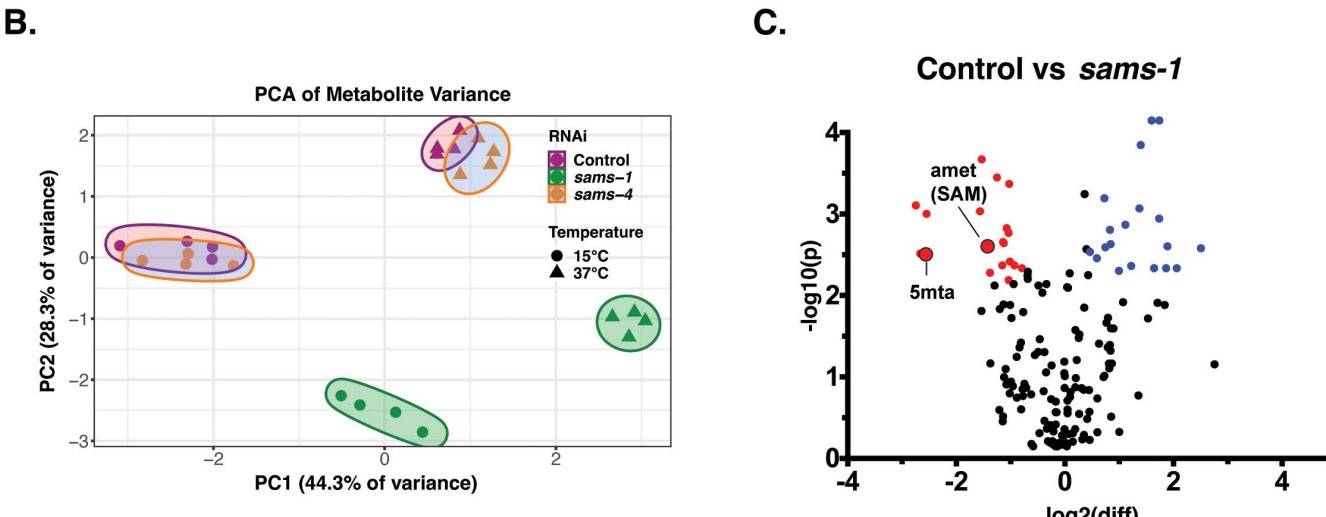

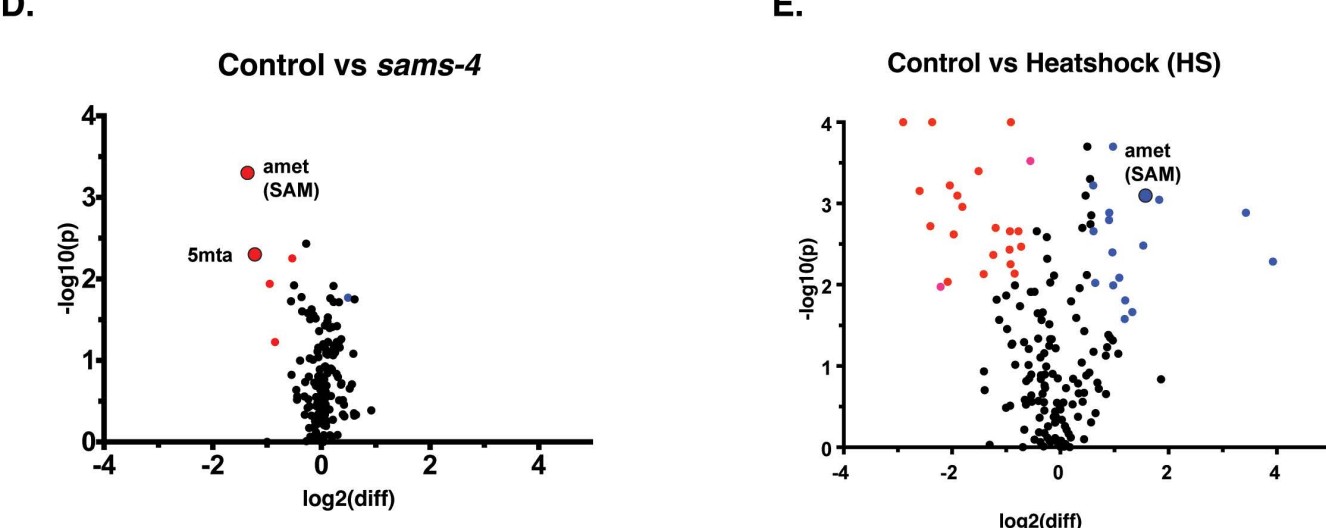

**Fig 1. Targeted metabolomics shows broad metabolic changes in *sams-1* animals in basal and heat shock conditions. (A)** Schematic of metabolic pathways using SAM. **(B)** PCA chart demonstrating distinct components in *sams-1* animals at control and heat shock conditions. Volcano plots comparing log10 fold changes in metabolite levels between Control and *sams-1(RNAi)* **(C)** or Control and *sams-4* **(D)** samples and Control vs. Heatshock **(E)**. Underlying data in S1 Table. 1-carbon cycle metabolites admet (SAM) and 5mta (methyladenosine) are highlighted.

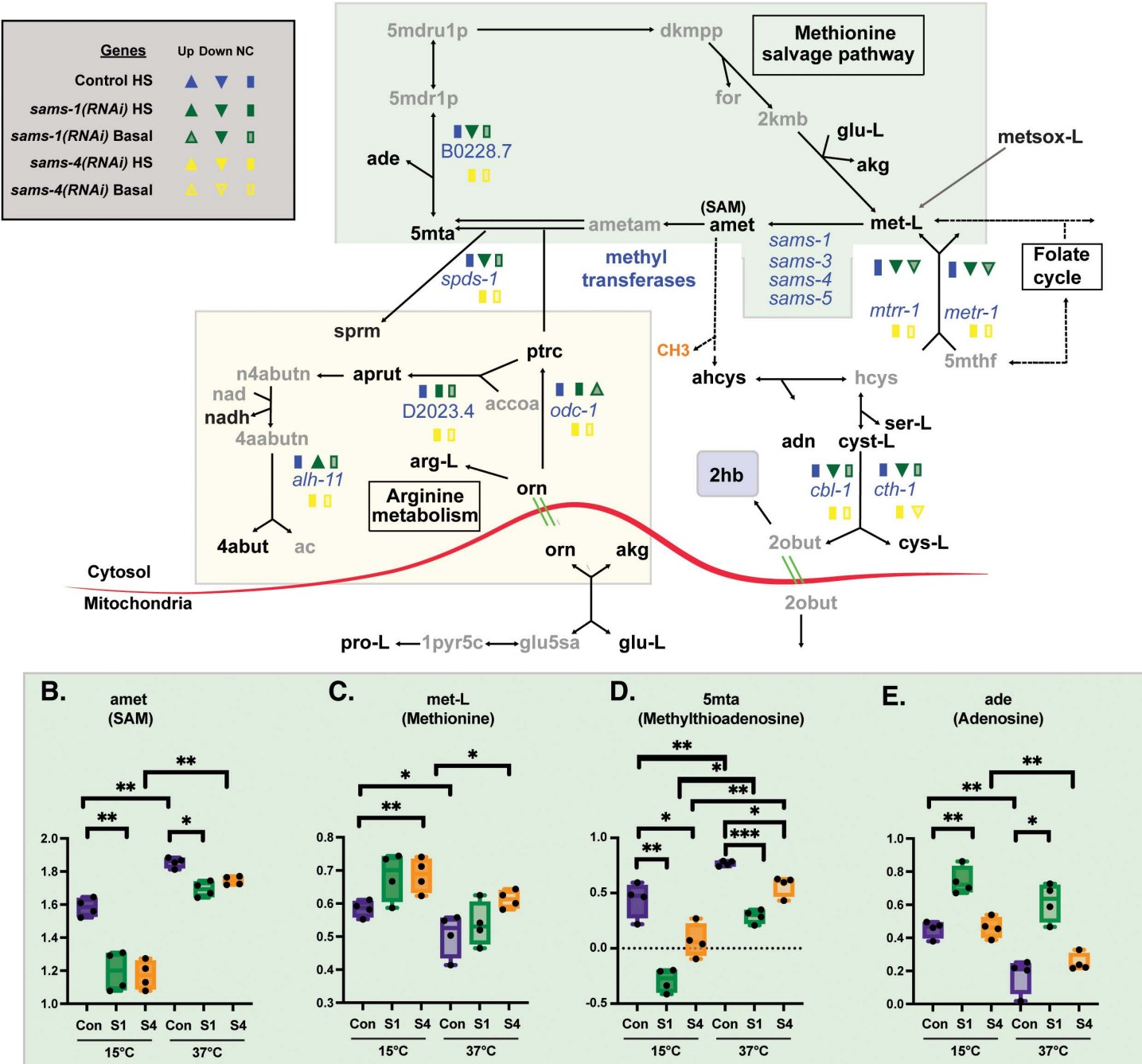

**Fig 2. Distinct effects on methionine salvage and polyamine pathways in *sams-1* and *sams-4(RNAi)* animals. (A)** Schematic derived from WormPaths [17] showing methionine salvage and arginine-derived metabolic pathways. See S1 Fig for schematic linking these pathways to mitochondrial fatty acid oxidation. Information regarding gene level and H3K4me3 status from Control, *sams-1* or *sams-4* animals in basal or heat-shocked conditions is marked in legend. Data is from Godbole and colleagues eLife 2023. Levels of selected metabolites from methionine salvage (green box) **(B–E)** are shown by box and whisker plots. Whiskers encompass data range and significance is determined by two-way repeated measures ANOVA. ns: $q$-value ≥ 0.05 *: $q$-value < 0.05, **: $q$-value < 0.01, ***: $q$-value < .001, ****: $q$-value < 0.0001. Underlying data is in S1 Table.

## SAM from SAMS-1 and SAMS-4 contributes to different metabolic pathways in heat stress

When control animals are exposed to heat stress there is a widespread change in metabolite profile (Fig 1E). Consistent with previous reports [5], SAM (amet) was one of the metabolites that was more abundant after heat stress in wildtype animals (Fig 1E and 2B). Notably, SAM increased dramatically in both *sams-1* and *sams-4(RNAi)* animals exposed to high temperature (Fig 2B), indicating that neither enzyme is uniquely required for the accumulation of SAM in heat stress. Because we are measuring steady-state levels of metabolites, we cannot definitively discern whether the increased levels of SAM observed in animals exposed to heat stress is caused by increased SAMS activity or if SAM use is restricted. However, we currently favor the interpretation that there is increased production of SAM by SAMS enzymes in heat stress because we observed a concomitant decrease in the substrate, methionine, in heat-stressed animals (Fig 2C). Moreover, the fact that SAM-dependent H3K4 methylation increases in heat stress argues against a global reduction in SAM use.

We mapped the metabolites onto WormPaths [17] to identify pathways that are most impacted when SAM synthesis is perturbed. As expected, depletion of *sams-1* and *sams-4* both impacted 1CC-related pathways in unstressed conditions - but with some notable differences. For example, S-adenosyl homocysteine (ahcys) was less abundant specifically in *sams-4(RNAi)* animals, which could suggest a decrease in SAM-dependent methylation reactions (S2A Fig). In contrast, levels of 5-methylthioadenosine (5mta) and adenosine (ade) were perturbed only in *sams-1(RNAi)* animals (Fig 2D). 5-methylthioadenosine (5mta) is generated when SAM is used in polyamine synthesis and adenosine (ade) is released [18]. In *sams-1(RNAi)* animals under both basal and heat-stress conditions, 5mta is low and adenosine (ade) is high (Fig 2D and 2E), suggesting increased methionine salvage. There was also evidence that polyamine synthesis is stimulated in *sams-1(RNAi)* animals. We found increased levels of polyamines such as putrescine (ptrc) and N-acetylputrescine (apurt) specifically in *sams-1(RNAi)* animals (S2B and S2C Fig), and in heat stress the abundance of ornithine (orn), a substrate for polyamine synthesis, was particularly sensitive to loss of *sams-1* (S2D Fig). These data suggest that SAM$_1$ contributes to at least two metabolic processes presented in our targeted metabolomics set, methionine salvage and polyamine synthesis. SAM$_4$, however, may be more important as a substrate for protein methyltransferase enzymes.

Depletion of *sams-1 or sams-4* also had different effects on metabolites of the transsulfuration pathway (Fig 2A), which removes homocysteine (hcys) from the methionine cycle to produce cysteine (cys-L). During heat stress, there is a dramatic decrease of 2-hydroxybutyrate (2hb) in wild-type animals (S2J Fig). This result could suggest that transsulfuration is reduced in heat shock or might reflect increased flux toward ketone body metabolism. RNAi of either *sams* gene in basal conditions results in similar decreases to each other in abundance of 2hb, possibly indicating a reduction in transsulfuration activity from decreased SAM production. Heat stress increases the level of 2hb in *sams-1(RNAi)* animals but not in *sams-4(RNAi)* animals (S2J Fig), perhaps suggesting that *sams-1* in particular is important to modulate flux through the transsulfuration pathway during heat stress.

## SAMS-1 promotes fatty acid oxidation and detoxification of propionyl-CoA

In addition to changes in methionine metabolism noted above (Fig 2), we found that *sams-1(RNAi)* had specific effects on metabolites associated with mitochondrial fatty acid β-oxidation (Fig 3A). In basal conditions, carnitine (crn), which is used to transport very long chain fatty acids into the mitochondria for β-oxidation, was elevated in *sams-1(RNAi)*, but not *sams-4(RNAi),* animals (S2K Fig). Heat stress resulted in a slight increase in carnitine in wild-type and *sams-4(RNAi)* animals, but *sams-1(RNAi)* animals had significantly higher levels even in basal conditions. These data suggest that β-oxidation is perturbed specifically in *sams-1(lof)* animals. The accumulation of carnitine may reflect decreased expression of carnitine palmitoyl transferase (cpt) enzymes in *sams-1(RNAi)* animals [5] or could be indirectly linked to other metabolic changes.

The abundance of several metabolites in the degradation of propionyl-CoA (ppcoa), a mitochondrial β-oxidation-like process, was also particularly sensitive to loss of SAM$_1$ (Fig 3A). Even in basal conditions, levels of 3-hydroxypropionate (3hpp) and 3-aminoisobutyrate (3aib) were lower in *sams-1(RNAi)* animals (Fig 3B and 3C). Upon heat stress,

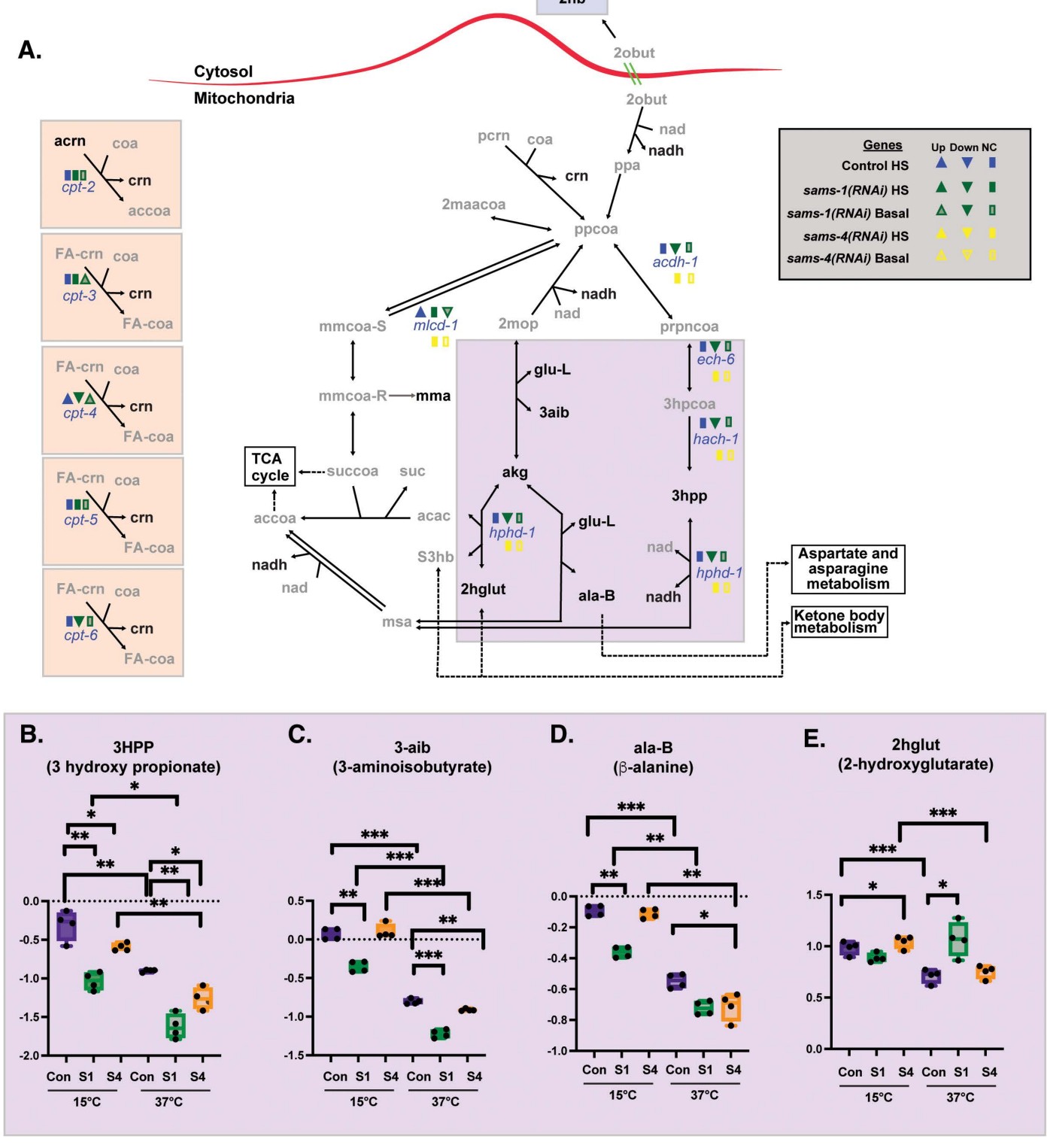

**Fig 3. Heat shock accentuates changes in multiple mitochondrial metabolites after *sams-1(RNAi)*. (A)** Schematic derived from WormPaths [17] showing methionine salvage and arginine-derived metabolic pathways. See S2 Fig for schematic linking these pathways to methionine salvage and arginine/polyamine synthesis pathways. Information regarding gene expression level and H3K4me3 status from Control, *sams-1* or *sams-4* animals in

basal or heat-shocked conditions is marked in legend, data is from [3]. Levels of selected metabolites from are shown by box and whisker plot and linked by color box to specific regions of the schematic **(B–E)**. Whiskers encompass data range and significance is determined by two-way repeated measures ANOVA. ns: $q$-value ≥ 0.05 *: $q$-value < 0.05, **: $q$-value < 0.01, ***: $q$-value < .001, ****: $q$-value < 0.0001. Underlying data is in S1 Table.

3-hydroxypropionate (3hpp) and 3-aminoisobutyrate (3aib) levels decline in all strains, exacerbating the depletion in *sams-1(RNAi)* animals. We observed a similar *sams-1*-associated depletion of β-alanine (ala-B) that was enhanced by heat stress (Fig 3D). The one exception to this pattern was the metabolite 2-hydroxyglutarate (2hglut), which was not depleted in *sams-1(RNAi)* animals in basal conditions (Fig 3E). With heat stress, 2-hydroxyglutarate (2hglut) levels decline in wild-type and *sams-4(RNAi)* animals, like other metabolites in this pathway, but did not change in *sams-1(RNAi)* animals (Fig 3E). This may reflect compensation by other metabolic pathways, such as isocitrate or ketone body metabolism that contribute to the steady state pools of 2hglut. Together, these data are in line with the use of SAM$_1$ to support catabolism of propionyl-CoA. This interpretation is consistent with studies from the Walhout lab that demonstrated *sams-1* impinged upon this pathway [19–21]. The expression of several genes that encode for enzymes in this pathway are reduced in *sams-1(RNAi)* animals (Fig 3A) [5], demonstrating a correlated change in gene expression and metabolite profile for this mitochondrial process.

## SAMS-1 is required to maintain age-associated expression of mitochondrial genes

Both reduction in *sams-1* and changes in mitochondrial function are associated with aging phenotypes [9,22]. To understand how gene expression changes in *sams-1* animals as they age, we measured mRNA abundance in WT and *sams-1(lof)* animals at day 1 (D1) and day 7 (D7) of adulthood by RNAseq (Fig 4A; S2 Table). PCA indicates that gene expression in *sams-1(lof)* animals is distinct from the wild-type control at both D1 and D7 (Fig 4B). This result is corroborated by hierarchical clustering analysis (Fig 4C), which shows that changes in transcripts are more similar in *sams-1(lof)* across age (D1 versus D7) than with age-matched wild-type controls. Scatter plots showed more variation between *sams-1(lof)* and wild-type controls at D1 than at D7 (Fig 4D and 4E). Relatively few of the genes that are upregulated in D1 *sams-1(lof)* animals are also upregulated at D7 (Fig 4F). In contrast, many genes downregulated at D1 in *sams-1(lof)* are further decreased in older animals at D7 (Fig 4G). This suggests that in addition to a core subset of genes that are downregulated with aging, *sams-1(lof)* animals acquire a distinctive gene expression signature. Our data set was consistent with previous studies [3] where loss of *sams-1* induced the upregulation of *fat-7* and decreased expression of *vit-2* (Fig 4H and 4I). Unrelated "housekeeping" transcripts, including *act-1* and *ama-2* (Fig 4J and 4K), were unaffected by genotype.

We used WormCat [23,24] to identify functional classes within the transcripts that changed in *sams-1(lof)* animals with age, focusing initially on down-regulated gene products (Fig 5A and 5B). In these data, some categories of genes, such as mRNA FUNCTION and CELL CYCLE, were enriched in both D1 and D7 (Fig 5A). This common set of genes that decline with age in both WT and *sams-1(lof)* animals age are likely to represent germline-specific gene sets. Since these changes occurred independently of *sams-1*, we did not further investigate these categories. We focused instead on categories that were uniquely enriched in the D7 data, where there were particularly noticeable changes in METABOLISM (Fig 5A). This category includes many nuclear-encoded mitochondrial genes, which is reflected in the enrichment of "mitochondrial" genes in the METABOLISM category. Inspection of the individual gene products in the METABOLISM: Mitochondria gene set show broad downregulation of mitochondrial genes with age in *sams-1(lof)* animals, particularly genes encoding components of the mitochondrial ribosome (Fig 5B–5F and S2 Table). While some genes encoding electron transport chain components were downregulated other pathways such as the citric acid cycle or ubiquinone synthesis were largely unaffected (S2 Table). We also found enrichment of genes included in the Gene Ontology groups GO:00005739 (Mitochondrion) (S3 Fig), which contains a broader array of genes encoding core mitochondrial-function proteins than the WormCat METABOLISM: Mitochondria set [23,24]. These GO sets included genes such as tRNAs (S2 Table) which could also be

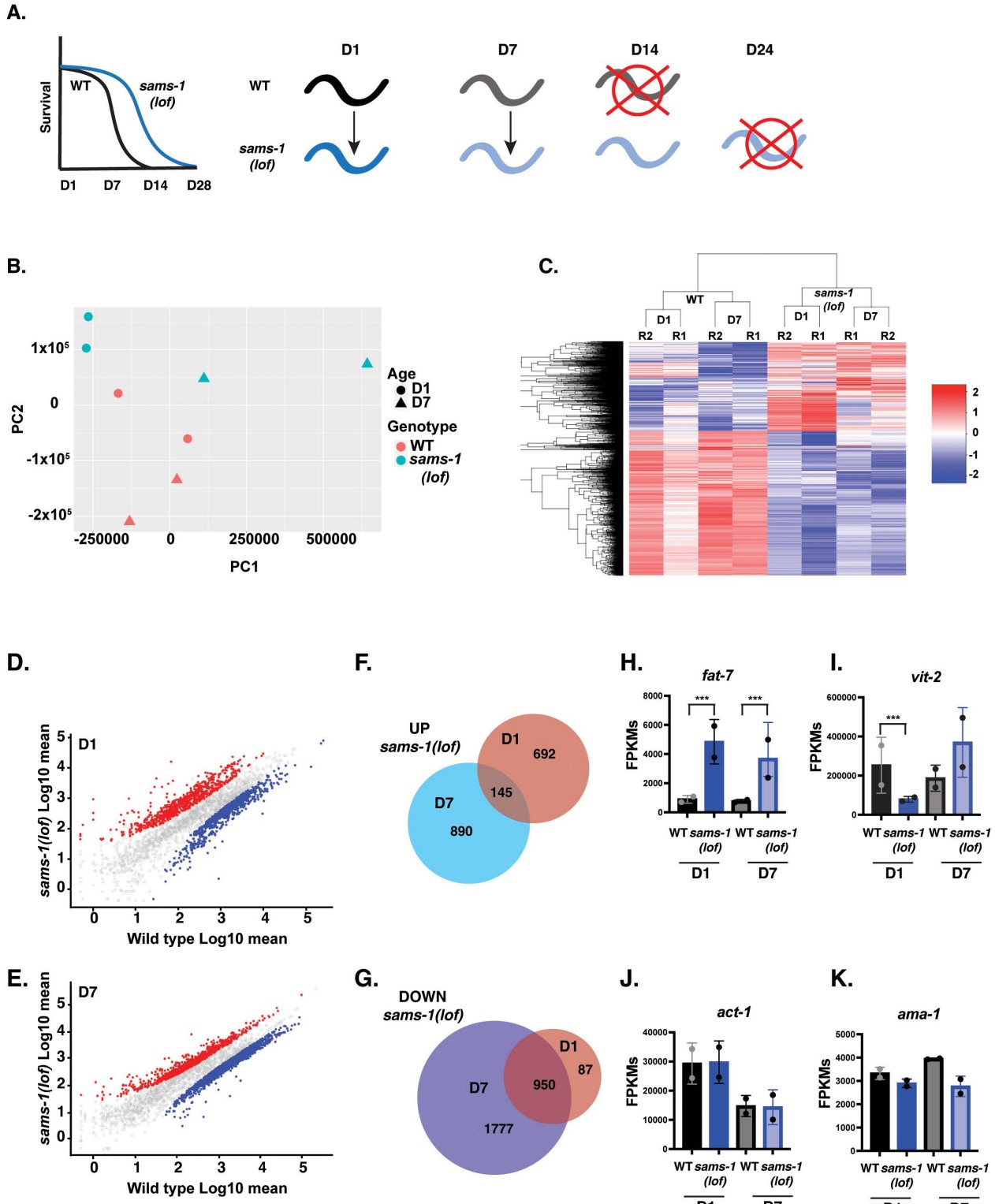

**Fig 4. Aging *sams-1(lof)* animals express distinct sets of genes from wild-type animals as they age. (A)** Schematic showing design of RNA seq experiment comparing younger and aged populations of long-lived *sams-1(lof)* animals. **(B)** Principal component analysis shows separation of WT and *sams-1(lof)* animals as they age for all genes. **(C)** Heatmap using k-means clustering demonstrating that gene expression patterns in wild-type and

*sams-1(lof)* animals cluster by genotype, rather than age. Scatter plots showing distribution of UP- and DOWN-regulated genes at D1 **(D)** and D7 **(E)**. Venn Diagram comparing overlap between genes up **(F)** or down **(G)** regulated at D1 vs. D7 in *sams-1(lof)* animals. Column graphs of FPKMs show two genes previously shown to be up- or downregulated after *sams-1(RNAi) fat-7* **(H)** and *vit-2* **(I)** [61] as well as two housekeeping genes, *act-1* (b-actin) and *ama-1* (RNA Pol II beta subunit) **(J, K)**. Whiskers encompass standard deviation and the *p*-adjust value calculated by Deseq2 shows significance including a false discovery rate with * *p* < 0.01, ** *p* < 0.005, *** *p* < 0.001. Underlying data is in S2 Table.

important for mitochondrial protein synthesis. We conclude that *sams-1* is required to maintain expression of mitochondrial genes as animals age.

Although our RNAseq data provided a strong indication of changes in nuclear encoded mitochondrial genes as *sams-1* animals age, this data contained only two replicates and limited grouping of the D7 *sams-1* replicates in the PCA, raising the possibility that false positives or negatives could affect our interpretations. Therefore, we obtained 3 independent biological replicates of each sample (S4 Fig, S3 Table, see S5A Fig for protocol schematic). This data set, where the *sams-1* groupings at D1 and D7 were clearly differentiated from each other, as well as wild type (S4A Fig), mirrored enrichment of mitochondrial genes in those downregulated in *sams-1* animals aged to D7 (S4B Fig). It also showed similar trends for individual nuclear encoded mitochondrial genes as our previous data set (S4C–S4F Fig), and expected upregulation of *fat-7* (S4G Fig) as in previous studies [8]. Finally, we confirmed these results using similar aged populations (S5A Fig) to isolate RNA for qRT-PCR and observed similar upregulation in the abundance of *fat-7* mRNA (S5B Fig) and downregulation of mitochondrial genes (S5C–S5E Fig).

### Limited overlap of *sams-1* upregulated genes with mitoUPR or autophagy regulators

WormCat analysis of genes with increased expression in *sams-1* animals revealed that category enrichment of lipid metabolic and stress-responsive gene expression was similar to our previous studies for the D1 animals [4,8]. This trend was enhanced, with enrichment in lipid metabolism and pathogen stress response categories increasing in the set of upregulated genes in D7 *sams-1(RNAi)* animals and demonstrating that these perturbations persist throughout the life span. We observed enrichment of some categories only in older D7 *sams-1(RNAi)* animals, such as neuronal function (subcategories synaptic function and neuropeptides) and transcription factors (Fig 5G).

Despite the downregulation of mitochondrial genes and lower levels of some mitochondrial metabolites (Figs 3 and 5), we did not observe transcriptional activation of the mito-UPR (mitochondrial unfolded protein response). The mito-UPR has been associated with increased life span and alterations in mitochondrial morphology [25]. There was no change in the abundance of *hsp-6* or *hsp-60* mRNAs in *sams-1(lof)* at D1 of adulthood in either set of our RNAseq data (Figs 5H, 5I, and S4H), consistent with previous microarray [3] and multiple RNAseq [4] experiments. Overlap between *sams-1* D1 upregulated genes and canonical mitoUPR sets [26,27] was limited (Fig 5J and 5K). However, this result does not agree with other reports showing that expression of P*hsp-6::gfp* is higher in *sams-1(RNAi)* animals [28–30]. or after depletion of a mitochondrial transporter proposed to import SAM into mitochondria, SLC-25A26 [29].

This GFP reporter has been shown to behave differently depending on if mitochondrial stress occurs during development compared to the induction of mitochondrial stress in adults [31].To ensure that our results were not confounded by loss of *sams-1* during development, we raised *sams-1(lof)* animals with supplemental choline, which restores PC levels in *sams-1* animals and rescues growth and SREBP-1-associated lipogenic defects [3,8,32,33]. We measured gene expression 24h after adults were removed (dropped) from the supplemental choline. Instead of mitoUPR activation, we observed a slight decrease in *hsp-6* (S5H Fig). However, there was a strong increase in expression of *fat-7* (S5G Fig), consistent with an acute loss of *sams-1* [3]. The discrepancy between our results and experiments using the P*hsp-6::gfp* reporter could suggest that the reporter does not faithfully reflect changes in expression of the endogenous gene product in all conditions. The P*hsp*-6::*gfp* strain has been shown to have increased stress sensitivity [34], suggesting that this multi-copy transgene may have neomorphic effects. We conclude that the transcriptional response of the mito-UPR is

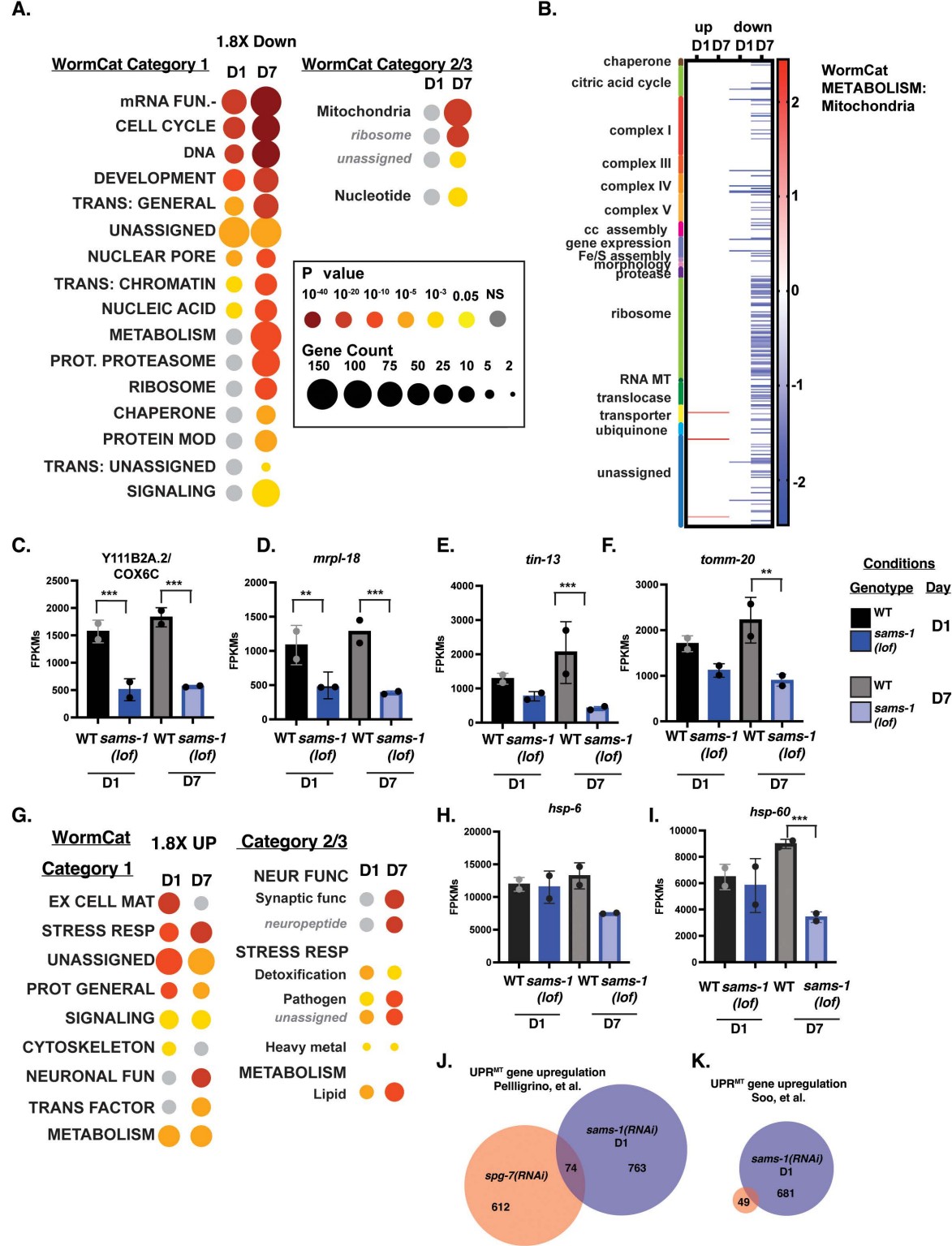

**Fig 5. Genes for mitochondrial-targeted proteins are downregulated in aged populations of *sams-1(lof)* animals. (A)** WormCat pathway category analysis of 2-fold downregulated genes in *sams-1(lof)* animals at D1 and D7 of adulthood. Heat maps showing gene expression patterns for all genes in the WormCat category **(B)** METABOLISM: Mitochondria with category 3 divisions are noted on the left axis **(C)**. Column graphs showing FPKMs from

D1 and D7 animals for nuclear encoded mitochondrial genes **(C–F)**. **(G)** WormCat pathway category analysis of 2-fold up genes. Box and whisker plots showing FPKMs from D1 and D7 animals for UPR^Mito genes **(H–I)**. Venn diagrams comparing sams-1 D1 upregulated genes with mitoUPR gene set in Pellegrino and colleagues **(J)** or Soo and colleagues **(K)**. Error bars show standard deviation and the *p*-adjust value calculated by Deseq2 shows significance including a false discovery rate with * $p < 0.01$, ** $p < 0.005$, *** $p < 0.001$. Underlying data is in S2 Table.

not activated in *sams-1(lof)* animals, and is therefore unlikely to contribute to observed life span and stress resistance phenotypes.

## Mitochondrial changes are specific to reduction in *sams-1*

Previous reports have linked changes in *sams-1* to mitochondrial fragmentation, which was proposed to result from loss of SAM-dependent epigenetic regulation [30]. Our finding of decreased mitochondrial metabolites and broad downregulation of mitochondrial genes in *sams-1(lof)* animals motivated us to more closely inspect the mitochondrial networks. We visualized the distribution of TOMM-20::mKate, a mitochondrially-localized fluorescent reporter [35] expressed in the intestine, in young (D1) and in older (D7) animals. In young *sams-1(RNAi)* animals we observed many smaller mitochondrial bodies as compared to the reticular network in wild-type controls (Fig 6A), similar to effects reported in [30]. This phenotype occurs in multiple tissues, as fission was also apparent in TOMM-20::mKate distribution in body wall muscle (Fig 6B).

We next asked how the mitochondrial network changes with age in *sams-1* animals. Consistent with reports that mitochondrial fission increases with age [36], we saw more fragmented mitochondria in the intestine and body-wall muscle in aging wild-type animals (Fig 6A and 6B). The situation was different in *sams-1* animals, which have fragmented mitochondria at D1. As *sams-1* animals aged, we instead noted a marked decrease in TOMM-20::mKate expression in both intestine and muscle tissues (Fig 6A and 6B). This result suggests an overall decrease in mitochondrial mass in aging *sams-1* animals. To evaluate this possibility, we measured the relative abundance of mitochondrial proteins, examined mitochondrial DNA levels, and compared mitochondrial footprints in imaging studies. We found that endogenous TOMM-20 protein levels were lower in D1 *sams-1(RNAi)* and further reduced at D7 (Fig 6C), consistent with the observed change in transgene expression. However, examination of an additional mitochondrial protein, the electron transport flavoprotein beta (*etfb-1*), did not show similar decreases. *etfb-1* mRNA levels, unlike *tomm-20*, were similar in aging or in *sams-1(lof)* animals (S2 Table). The nuclear to mitochondrial genome ratio was the same in *sams-1(RNAi)* animals and wild-type controls in D1 adults (Fig 6D).

Next, to determine if mitochondrial effects were specific to loss of *sams-1*, or were a general effect of low SAM, we compared loss of each *sams* gene using TMRE (Tetramethylrhodamine ethyl ester) staining to compare membrane potential, quantitate mitochondrial networking, and evaluate mitochondrial footprints. Imaging of TMRE in the hypodermal tissue confirmed loss of mitochondrial networks in *sams-1*, but not *sams-4* or *-5* animals (Fig 6E). Quantitation of junction points by MITOMAPR [37] shows that while networking and mitochondrial footprints are markedly reduced in *sams-1* animals, TMRE intensity, a measure of membrane potential, is not changed (Fig 6F–6H). We conclude that mitochondrial fragmentation occurs specifically when *sams-1* is depleted, and that the morphological changes are specific to reduction in *sams-1* and occur early in adulthood, at the time of increased stress resistance.

TMRE is a lipophilic dye and levels may not always predict changes in mitochondrial function [38]. Direct measurement of oxygen consumption rate (OCR) is a more quantitative method and *C. elegans*-specific protocols have been developed for the Seahorse Bioanalyzer [39,40]. However, because whole worms are measured, size differences may also confound results [41]. Because the small size and diminished germline (a mitochondria-rich tissue) of *sams-1* animals would make normalization of whole animal assays difficult, we chose to compare OCR between intestinal mitochondria that had been isolated from control or *sams-1(RNAi)* animals through IP from a *ges-1*::TOMM-20::mKATE::HA expressing strain (Fig 6I). Isolated mitochondria were supplied with succinate, pyruvate, glutamate, and ADP (adenosine triphosphate). Mitochondria

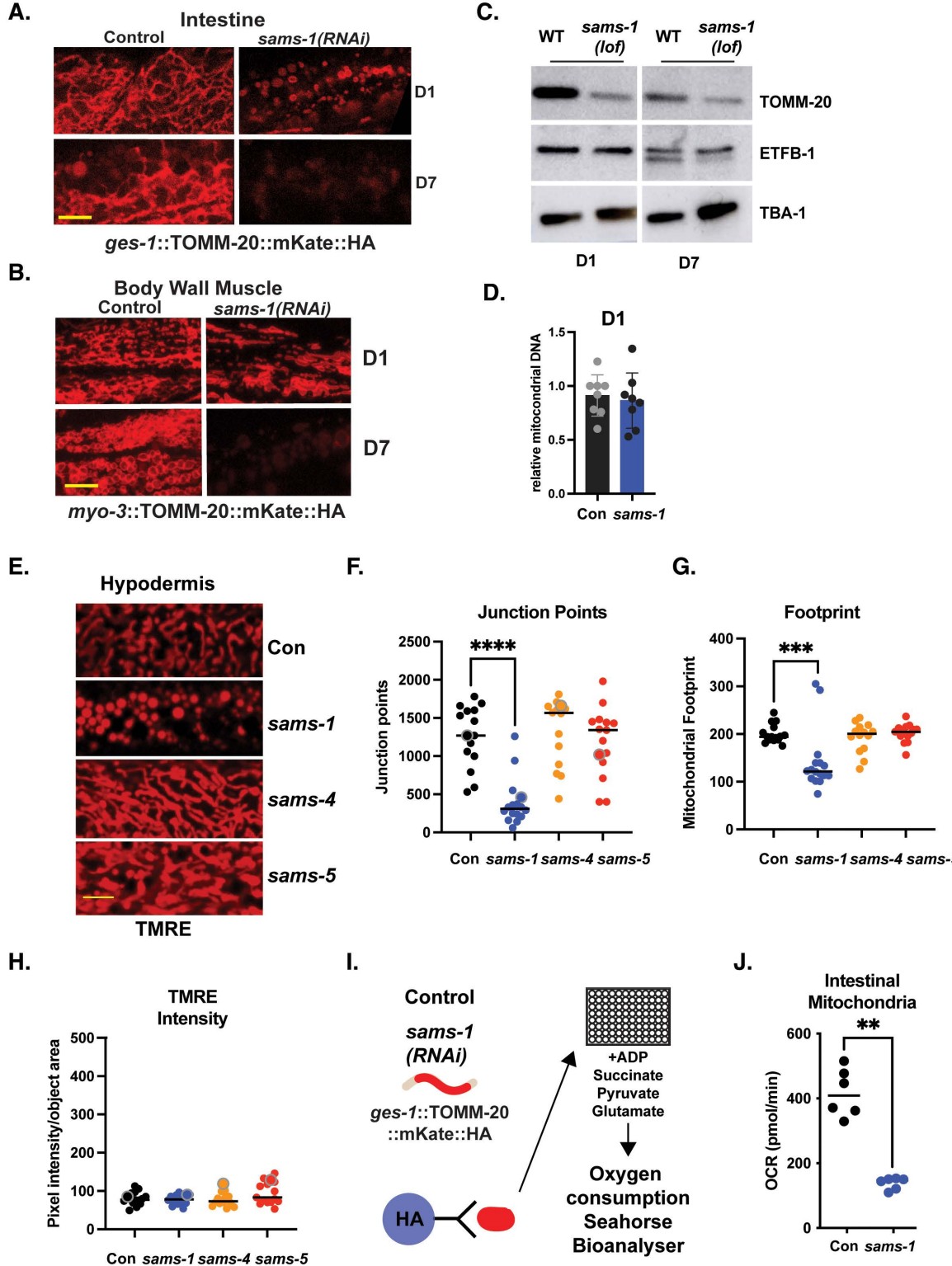

**Fig 6. *sams-1* knockdown has specific effects on mitochondrial morphology.** *sams-1(RNAi)* causes changes in mitochondrial morphology at D1 of adulthood and loss of TOMM-20::mKate by D7 in intestinal **(A)** and body wall muscle cells **(B)**. **(C)** Immunoblot comparing levels of endogenous TOMM-20 along with ETFB-1 in wild-type and *sams-(lof)* animals as they age using alpha tubulin (*tba-1*) as a loading control. Raw images are

in S1_raw_images.pdf. **(D)** Q-PCR comparing mitochondrial and nuclear DNA levels in D1 control and *sams-1(RNAi)* animals. **(E)** Projections from spinning disc confocal images of TMRE staining of D1 adult *C. elegans* after RNAi of each SAM synthase shows that mitochondrial effects are specific to *sams-1*. Quantitation of networking (junction points) is in **(F)** with TMRE footprint in **(G)** and intensity in **(H)**. Schematic **(I)** and OCR from representative Seahorse assay in mitochondria purified from Control or *sams-1(RNAi)* animals. Significance calculated by the Mann–Whitney test is shown with * $p < 0.01$, ** $p < 0.005$, *** $p < 0.001$. Scale bar is 10 microns. The data for Fig 6F–6H, Fig 7C, 7D and Fig S7A were collected at the same time; thus, Control and *sams-1(RNAi)* graphs represent the same data set. Distinct images were chosen for panels marked within quantitation with large, contrasting point. The underlying data is in S1 Data.

from control animals showed efficient basal respiration in these fuel conditions, while equivalent concentrations of *sams-1* mitochondria had lower OCRs. Thus, *sams-1* mitochondria are less able to utilize TCA intermediates as fuel, showing functional as well as morphological differences from SAM depletion.

### Insufficient production of phosphatidylcholine leads to changes in mitochondrial morphology

We hypothesized that the increased stress resistance in *sams-1(lof)* animals may be related to *sams-1*-specific metabolic changes. Membrane lipids are critical for mitochondrial morphology and SAM synthases have an important role in PC production [42]. SAM is required for the methylation of PE to form PC in mammals [42]. In *C. elegans*, the PMT-1/PMT-2 phosphoethanolamine methyltransferases act to methylate phosphoethanolamine (P-Eth) to generate phosphocholine (P-Cho), which can then be used to produce PC [33] (schematic in Fig 7A). LCMS assays showed decreases in PC in *sams-1(RNAi)* animals (Fig 7B; S3 Table), corroborating our previous results [3,8,43], and were equivalent to reduction in *pcyt-1*, the rate-limiting enzyme for the de novo synthesis of PC (Fig 7A). PC levels in *sams-4(RNAi)* animals were the same as wild-type, indicating that $SAM_4$ is not a major substrate for PMT-1/PMT-2 methyltransferases. PC metabolism is tightly linked to stored fats (triglycerols), diacylglycerols, and phosphatidylglycerols (PGs); these all change in *sams-1* animals (S6A–S6D Fig; S4 Table) consistent with our previous studies [43,44]. In contrast, loss of *sams-4* had little effect on any lipid class (S6A–S6D Fig; S4 Table). We conclude that SAMS-1 is uniquely important for SAM-dependent lipid metabolism.

The fact that PC decreases in *sams-1* animals, where we observe fragmented mitochondria (Fig 6A, 6B, and 6E), but not in *sams-4* animals which have normal mitochondrial morphology (Fig 6E), suggested that reduced PC may be the underlying cause of mitochondrial fragmentation. We first grew *sams-1(lof)* animals with supplemental dietary choline, which restores PC levels in *sams-1(lof)* animals [3,8,32,33]. Dietary choline fully suppressed mitochondrial fragmentation in *sams-1(RNAi)* animals (Fig 7C and 7D). Supplemental choline was associated with increased TMRE intensity in both control and *sams-1(RNAi)* animals (S7A Fig). Next, we asked if other genetic manipulations that reduce PC synthesis also induced mitochondrial fragmentation. We found that mitochondria fragmented after knockdown of either *pmt-2* (Fig 7E and 7F), the methyltransferase required to make PC from PE, or *pcyt-1* (Fig 7G and 7H), which is required for dietary choline to be incorporated into PC. The intensity of TRME was reduced moderately in *pmt-2(RNAi)* animals (S7B Fig), which induces a more severe growth restriction than depletion of *sams-1* [8,33], but was not changed in animals with reduced *pcyt-1* (S7C Fig). These data strongly support our assertion that the fragmentation of mitochondria in *sams-1* animals results from decreased PC production.

PC, the most abundant phospholipid in cellular membranes, is synthesized in the ER and must be transported to other cellular organelles, including mitochondria [42]. Decreases in PC can affect membrane properties [45], limiting membrane curvature or affecting function of proteins within membranes [46]. PC also acts as a substrate for phospholipases, such as Phospholipase D, and therefore could indirectly impact mitochondria through signaling functions. To ask if PC levels had potentially direct effects on mitochondrial morphology, we identified *C. elegans* orthologs of proteins that could be involved with PC transport into the mitochondria (VPS-13D) or act within the mitochondria to transfer PC to the inner membrane (STRT-7/STARD7). *vps-13D* encodes a lipid transfer protein that can facilitate movement of PC from the ER to the mitochondria; reduction in function causes fission and increased entry into mitophagy pathways in yeast, *Drosophila*, and

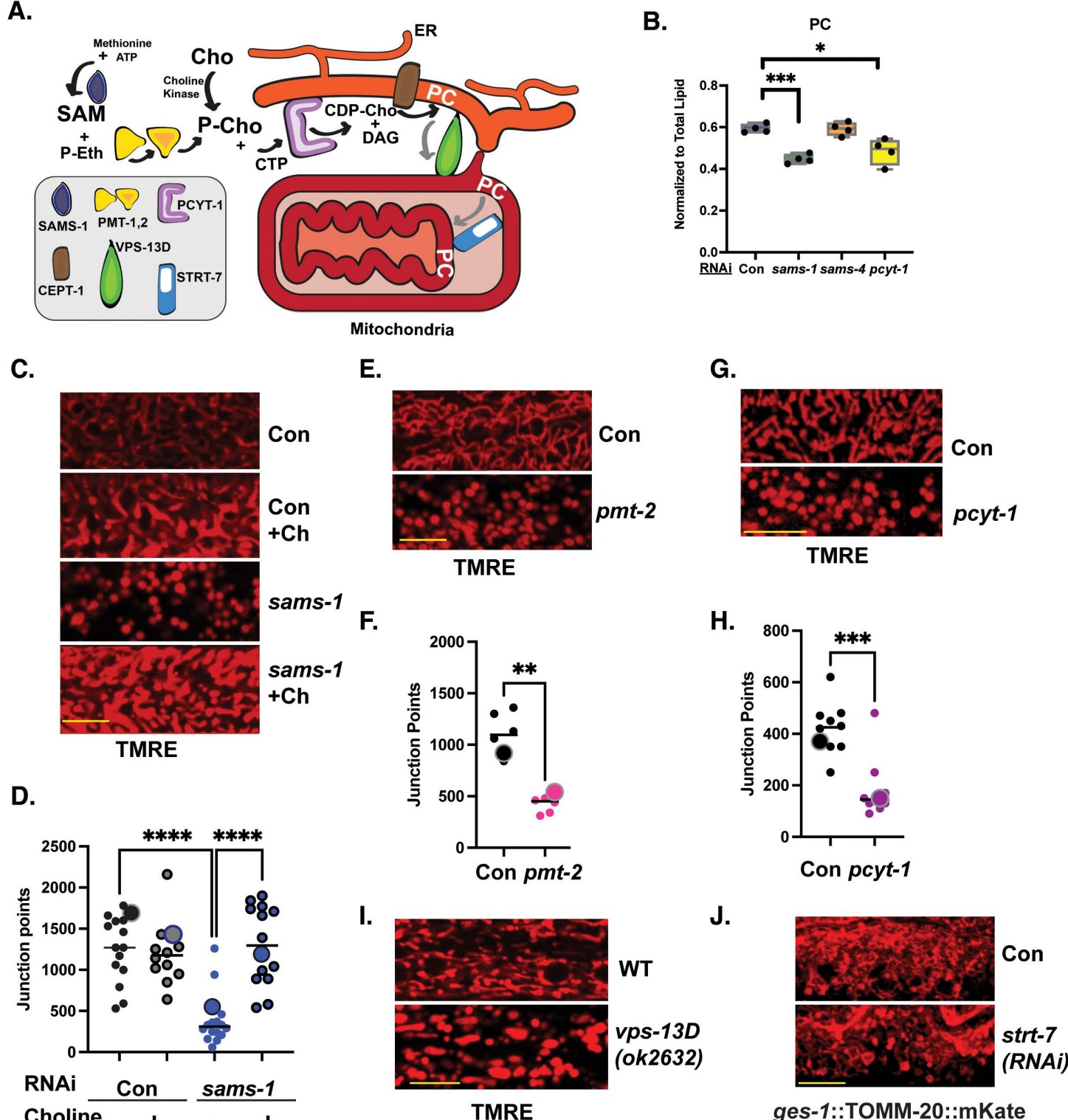

**Fig 7. Changes in mitochondrial morphology in *sams-1* animals are linked to low PC. (A)** Schematic showing connections between SAM and PC synthesis to trafficking of PC to and within the mitochondria. **(B)** LCMS analysis shows that PC synthesis defects in low SAM are specific to *sams-1* (See also S6 Fig, underlying data is in S4 Table). **(C)** Rescue of PC synthesis with dietary choline restores mitochondrial networks in D1 *sams-1(RNAi)*

animals; network quantitation of TMRE intensity is in **D** with TMRE intensity in Fig S6A. RNAi of the PC methyltransferase *pmt-2* (**E, F**, S7B Fig), the rate-limiting enzyme for PC production *(pcyt-1)* (**G, H**, S7C Fig), or the ER-mitochondrial phospholipid transfer protein *vps-13D* (**I**) induces mitochondrial fission, visualized by TMRE staining, similar to reduction after *sams-1* RNAi. **(J)** Spinning disc projection of intestinal cells expressing TOMM-20::mKate to visualize changes in the mitochondrial gut after RNAi of the mitochondrial phospholipid flippase *strt-7*/STAR7. Quantitation in S7DE Fig. Images are spinning disc projections except for **G**, which was taken on a point scanning confocal. Significance calculated by the Mann–Whitney test is shown with * *p* < 0.01, ** *p* < 0.005, *** *p* < 0.001. Scale bar is 10 microns. The data for Fig 6F–6H and Figs 6C, 6D/S7A were collected at the same time; thus, Control *and sams-1(RNAi)* graphs represent the same data set. Distinct images were chosen for panels marked within quantitation for each panel with large, contrasting point. The underlying data for 7F, 7H, 7D is in S1 Data.

humans [47–51] and changes in mitochondrial morphology in *C. elegans* [52]. Orthologs in humans are associated with movement disorders [53], which are also linked to mitochondrial functions [54].

Consistent with the loss of mitochondrial networks in body wall muscle observed by Yin and colleagues, we found that *vps-13(ok2190)* animals lost network junctions after TMRE staining of hypodermal tissues (Fig 7I) [52]. PC transport can also be regulated within these organelles by mitochondrial-targeted isoforms STARD7 [55–58]; cells lacking STARD7 have lower PC within mitochondria, exhibit fission of mitochondrial networks with increased mitophagy and abnormal cristae [55,56,59]. We identified two STAR Domain containing proteins in the *C. elegans* genome that were orthologous to STARD7 and STARD10 respectively (S6I and S6J Fig). Like *STARD7*, *strt-7* contains a mitochondrial targeting sequence and is predicted by Wormbase [60] to have a predominantly intestinal expression. RNAi knockdown of *strt-7* also disrupted the mitochondrial network (Figs 7J and S7D). Taken together, our data support the idea that changes in mitochondrial morphology in *sams-1* animals are driven by decreased mitochondrial PC which lead to fragmentation of the mitochondrial network.

## Fragmented mitochondria in *sams-1* animals are targeted to autophagosomes

Changes in mitochondrial morphology are often associated with cellular stress [61]. For example, mitochondria fragment in *C. elegans* larvae exposed to heat stress [62], see also S8A Fig). Depleting *drp-1*, the dynamin family GTPase required for mitochondrial fission, prevents heat-stress-induced mitochondrial fragmentation and renders animals more sensitive to thermal stress [62,63]. This suggests that regulated changes in mitochondrial morphology may contribute to stress survival. Conversely, animals with fragmented mitochondria due to loss of *fzo-1*, the *C. elegans* orthologue of mitofusin [62,63], are more resistant to heat stress. Fission allows mitochondria to more easily enter the autophagy pathway and be turned over through mitophagy [63,64], which can help clear cellular material damaged by stress and enhance survival.

We used epistasis analysis to ask if the changes in morphology of the mitochondria network that we observed when PC production was limited acting through the cellular machinery that balances mitochondrial fission and fusion. As in previous studies [62,63], we observed that increased mitochondrial fragmentation after depletion of *fzo-1* increased survival, while loss of *drp-1*, which leads to more fused mitochondria, impairs stress resistance (Fig 8A). However, survival of *sams*-1 animals does not meaningfully change after depletion of either *fzo-1* or *drp-1* (Fig 8B). Wei and colleagues have shown that depletion of *sams-1* RNAi does not fully restore mitochondrial morphology in *drp-1(lof)* animals [30], and we found that mitochondrial fragmentation in *sams-1(lof)* animals is not exacerbated by depletion of *fzo-1* (S8B and S8C Fig). Lowering PC within membranes reduces membrane bending [45], and may cause organelles to form larger, rounded shapes to minimize complex curvature [65]. Taken together, these results suggest that the membrane constraints in low PC leads to disrupted networks, that like *fzo-1* RNAi, increase stress resistance but are refractory to additional modulation by the fission or fusion machinery.

We next sought to directly examine the effects of reduced PC production on mitochondria using high-pressure freeze fixation/transmission electron microscopy (TEM). Whereas intestinal mitochondria in control animals where elongated with visible cristae and were dispersed throughout the cytoplasm, we noted that the mitochondria of *sams-1(RNAi)* animals were preferentially distributed along the intestinal lumen, had a rounded or bloated morphology, and barely discernible

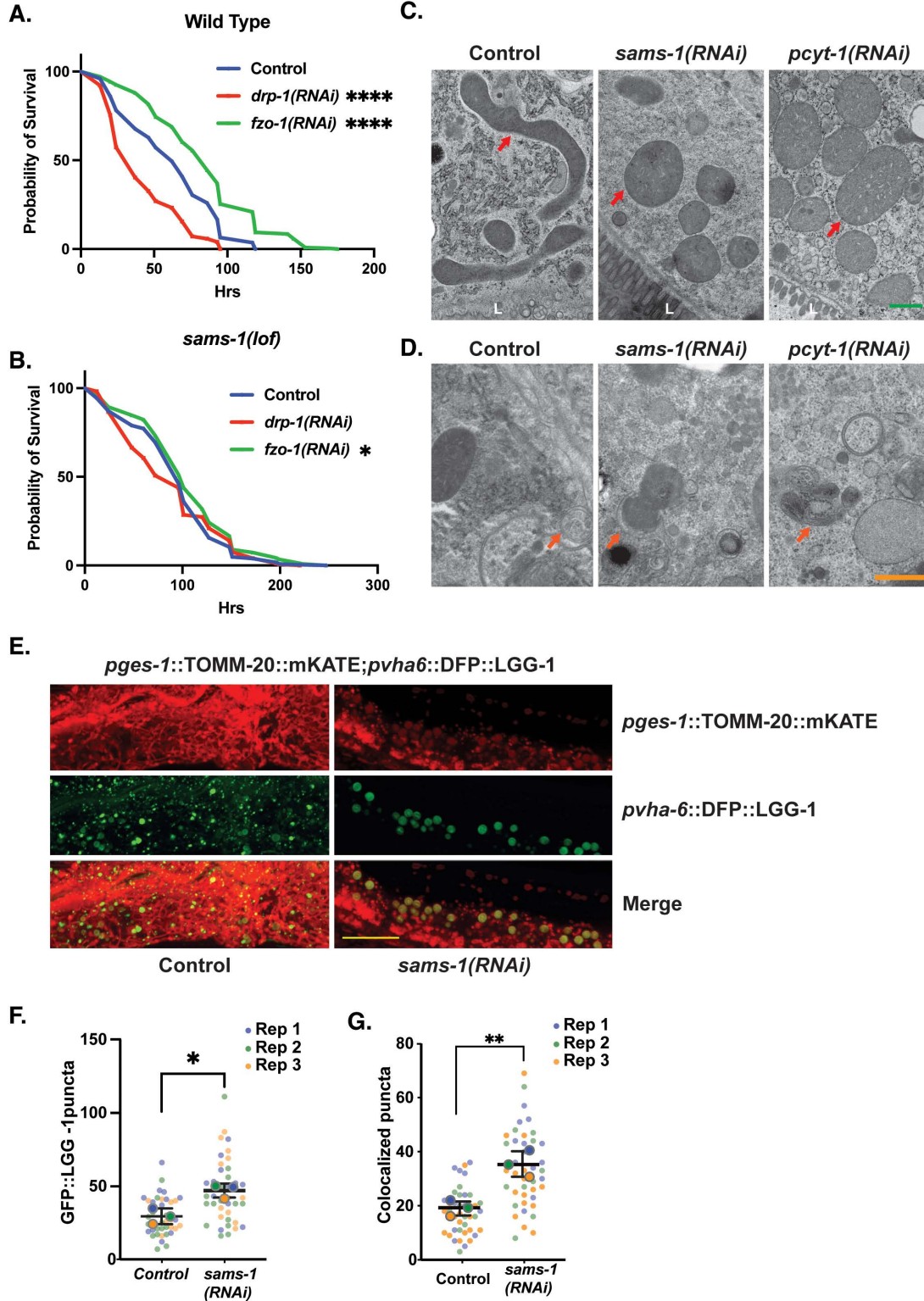

**Fig 8. Increased mitophagy in *sams-1(RNAi)* animals links changes in PC to heat shock survival.** Average of three survival assays comparing effects of heat shock on wild-type **(A)** or *sams-1(lof)* **(B)** animals undergoing RNAi for *drp-1* or *fzo-1*. Panel A confirms results from [63]. TEM images of Control, *sams-1* or *pcyt-1* RNAi animals comparing mitochondrial morphology, localization and localization within intestinal cells **(C)** and showing

examples of mitochondria in proximity to double membrane autophagosome-like structures **(D)**. Scale bar is 10 microns. **(E)** Confocal projections obtained on a Nikon Spinning disc of intestines from *sams-1(RNAi)* animals showing increased numbers of DFP::LGG-1 autophagosome puncta and overlap with TOMM-20::mKate marked mitochondria. Scale bar is 500 nm. Quantification of DFP::LGG-1 puncta is in **(F)** and of overlap between DFP::LGG-1 puncta and TOMM-20::mKate in **(G)**. Significance calculated by the Mann–Whitney test is shown with * $p < 0.01$, ** $p < 0.005$, *** $p < 0.001$. The underlying data is in S1 Data.

cristae structure (Figs 8C and S8D). We observed a similar effect in *pcyt-1(RNAi)* animals (Figs 8C and S8D). We also noted tubulation of ER membranes in addition to the mitochondrial effects, which is not unexpected as PC is a major constituent of most cellular membranes. These results further support our assertion that *sams-1* mitochondrial phenotypes result from depletion of PC.

Mitochondrial fragmentation can be the first step in the specific degradation of mitochondrial components by autophagy, or mitophagy [66]. We noticed that fragmented mitochondria in *sams-1(RNAi)* and *pcyt-1(RNAi)* animals were often observed near cupped-shaped double membrane structures (Fig 8D), suggesting that they may be subject to mitophagy. Previous studies have shown that *sams-1(ok2946)* animals have more GFP::LGG-1 puncta in larval hypodermal, which mark forming autophagosomes, and that these animals require *bec-1*, a key autophagy mediator, for full life span extension [67]. We compared the effects of depleting *sams-1* or *sams-4* on autophagic flux using a strain with two fluorescent proteins that can distinguish between forming autophagosomes and autolysosomes [68]. We found more fluorescent puncta in *sams-1(lof)* animals (S8E and S8F Fig), corroborating previous studies. The amount of GFP::LGG-1 was decreased relative to mKate, suggesting that flux through the autophagy pathway was increased. Loss of *sams-4* had no effect on autophagy flux using this assay.

To determine if the mitochondrial fragments produced in *sams-1(lof)* animals were removed by mitophagy, as is the case when mitochondrial PC is lowered by loss of VPS13D or STARD7 [50,56], we knocked down *sams-1* in a strain where intestinal autophagosomes were marked with DFP::LGG-1 and mitochondria with TOMM-20::mKate [69]. In *sams-1(RNAi)* animals, most mitochondria with bright TOMM-20::mKate were clustered near the basal side of the intestinal cells (Fig 8E), consistent with our TEM images (Figs 8C and S8D). However, we also noted that there were lighter, less refractory puncta closer to the apical side (Fig 8E). The number of DFP::LGG-1 puncta was higher in *sams-1(RNAi)* animals (Fig 8F), consistent with previous results [67]. Notably, we found that DFP::LGG-1 puncta colocalized with mitochondria in *sams-1(RNAi)* animals (Fig 8E and 8G), suggesting that these mitochondria could be primed to enter mitophagy pathways, allowing clearance and increasing stress resistance.

An alternative hypothesis for the increase in autophagic puncta and overlap with mitochondria in *sams-1* animals could be transcription-mediated effects on autophagy genes as reported in Lim and colleagues. In this study, expression of several autophagy-linked genes were increased by slight, but statistically significant margins, measured by qRT-PCR in a *sams-1* deletion allele [67]. These changes were linked to H3K4-mediated control of the master regulator of autophagy, *hlh-30*/TFEB along with *pha-4*/FOXA1 as a mechanism to explain increases in autophagy in low SAM. However, we have not observed increased expression of autophagy regulators after loss of *sams-1* in multiple independent RNAseq and microarray experiments [3–5]. Consistent with this, there was no consistent upregulation of the autophagy regulators *pha-4* or *hlh-30* in either of the RNAseq data sets we collected in this study (S9 Fig; S2 and S3 Tables). In our new data sets, only *sqst-1* responded transcriptionally to loss of *sams-1* in our RNAseq studies (S9C Fig; S2 and S3 Tables). Finally, although sequencing of our previous Cut&TAG studies [5] identified H3K4me3 peaks in the same regions analyzed by ChIP-PCR in [67], these were not differentially regulated (S10 Fig; S2 and S3 Tables), suggesting that H3K4me3-mediated control of the HLH-30 response may not link SAMS-1 to autophagy in all contexts. However, there may be other stress responses activated by loss of *sams-1*, as we did note an enrichment in STRESS RESPONSE categories as *sams-1* animals aged. This category included multiple transcriptional factors that may contribute to stress responsive gene expression, such as the AP1 components *fos-1* and *jun-1* and nuclear hormone receptor *daf-12* (S9G–S9I Fig). Thus, a

PLOS Biology

more generalized transcriptional stress response may be linked to the lowering of PC and changes in mitochondria or other membrane organelles in *sams-1* animals.

## Discussion

We have found that different SAM synthase enzymes have distinct effects on cellular metabolic processes. We show that SAMS-1 contributes to a variety of metabolic pathways, including a mitochondrial β-oxidation-like pathway and the synthesis of the phospholipid PC. SAMS-4 has a more restricted role, perhaps serving primarily as a substrate for a more limited set of histone or other protein methylation reactions. These results support the idea that SAM generated by different SAMS enzymes is used for specific cellular and metabolic processes. This may explain our previous observations that different SAM synthases have distinct effects on life span, heat stress survival, gene expression profiles, and H3K4me3 patterns [4,5]. More generally, our study highlights the importance of considering synthase-specific effects when determining how low SAM may exert phenotypic effects.

Our studies reveal that a key function of SAMS-1 is to generate PC, and that this is essential to maintain mitochondrial networks in young adults. Furthermore, *sams-1* is important to support expression of many nuclear encoded mitochondrial genes in older animals. In *sams-1(lof)* animals, reduced PC production leads to mitochondrial fragmentation, changes in OCR and degradation by mitophagy (model in S8G Fig). We measured OCR in purified mitochondria supplied with TCA cycle intermediates, however, fission can produce mitochondrial which may be more efficient with different fuels, such as those produced by beta oxidation [70,71]. The mitochondrial functions of SAMS-1 may be conserved. In mammals, loss of MAT1A in liver causes mitochondrial dysfunction associated with changes occurring in alcoholic fatty liver disease [72], and mutations in choline kinase, which mediates an initial step in PC synthesis, are associated with changes in mitochondrial morphology and mitophagy in mouse muscle tissue [73] and multiple human neurological syndromes [53,54,74]

PC is a major component of most cellular membranes, and is also used in the synthesis of other complex lipids such as cardiolipin (CL), which plays an important role in mitochondrial membranes [75]. However, there is no change in the abundance of CL in *sams-1* animals, arguing against a major role for CL in mitochondrial fragmentation from knockdown of *sams-1*. There were changes in the abundance of other lipids linked to mitochondrial function in *sams-1(RNAi)* animals, including PG, SPH, CoQ, and Acyl Carnitines (S6D–S6H Fig; S4 Table). These changes may be a compensatory response to decreased PC abundance or altered mitochondrial function, as we observed similar changes in *pcyt-1(RNAi)* animals.

We also noted changes in several metabolites of the propionate shunt, a β-oxidation-like pathway. This validates our previous report that showed genes in this pathway lost H3K4me3 and heat-shock-associated gene expression in *sams-1(lof)* animals [5]. There are clear links between the regeneration of methionine and SAM synthesis through vitamin B12 and the detoxification of hydroxy propionate (3HPP) [19,20,76,77]. Recent studies have linked increased 3HPP with defects in mitochondrial morphology including sheet-like structures and vacuoles [78]. These can be rescued by B12 or the low-branched chain amino acid (BCAA) fatty acid diet of the *Escherichia coli* strain used for *C. elegans* RNAi studies. While B12 and BCAA metabolism can affect the 1CC, *sams-1(RNAi)* animals showed no significant differences in the BCAAs leucine or isoleucine, although valine levels dropped. The metabolomics in this study, combined with our previous chromatin and gene expression studies [5], suggest this pathway is modulated mostly through effects on H3K4me3 and expression in genes for metabolic enzymes in this pathway. Polyamines may also affect mitochondrial morphology and mitophagy [79], however, in our experiments the levels of metabolites in the polyamine or arginine biosynthetic pathways did not co-vary with *sams-1* effects on heat shock, thus spermidine appears able to induce mitochondrial stress. However, interpreting negative effects from a small list of steady state metabolites is difficult and measurement of flux from methionine to SAM and downstream metabolites is necessary to fully evaluate this hypothesis.

Reduced PC abundance can affect mitochondrial morphology and membrane potential through effects on membrane dynamics [75,80]. PC plays a critical role in membrane stability and curvature, as its cylindrical structure promotes bilayer

formation [45]. The effect of decreased PC on membrane dynamics is well studied in lipid droplet biology [81], as PC depleted membranes are less amenable to curvature, with larger and more regular formation of lipid droplets. This is consistent with the larger, less networked mitochondria we observed in both the *sams-1* and *pcyt-1* RNAi animals. In mitochondria, low PC may also favor fission rather than maintenance of complex fused mitochondrial networks and alter modifications of mitophagy regulators such as Atg8 [82] or Atg44 [64], which could influence incorporation of damaged mitochondria into the autophagy machinery. There may be even deeper links between PC and autophagy/mitophagy, as Sutter and Tu [83] identified ChoP (choline phosphotransferase) in a screen in yeast for regulators of autophagy/mitophagy induced by low methionine.

The relative amount of PC in the membrane varies between cellular organelles, with highest levels in the ER, followed by the Golgi, mitochondria and lysosomes, with the lowest proportion in the plasma membrane [84]. It could be, therefore, that decreased PC has distinct effects on different organelles. For example, we previously showed that low PC blocks ARF-1 cycling at the Golgi in *C. elegans* and mammalian cells [44], and studies from the Spang lab show that ARF-1/ARF1 impacts mitochondrial dynamics in *C. elegans*, yeast, and mammalian cells [85,86]. This raises the possibility that low PC blocks ARF-1 cycling at the Golgi and this drives mitochondrial fragmentation. However, loss of *arf-1* does not result in mitochondrial fragmentation [86], as in *sams-1* animals. VPS13D and STARD7 can also affect distinct organelles or membranes in mammalian cells, although the direct targeting of some isoforms to the mitochondria or ER-mitochondrial contact sites supports the simplest, most direct explanation [47,87] for phenotypic similarities. Moreover, loss of STARD7, which causes mitochondrial dysfunction associated with a seizure disorder in humans [55], has been shown to alter mitochondrial morphology with increased blebbing and loss of cristae structure in mammalian cells [55,59]. And, similar to our observations in *sams-1(lof)* animals, the change in mitochondrial morphology resulting from loss of STARD7 is independent of dynamin-related GTPases [59] and associated with increased mitophagic flux in C1C12 muscle cells [56].

Reduction in SAM has broad effects on metabolite levels and methylation patterns leading to diverse phenotypes. Our study highlights the importance of demonstrating synthase specific downstream roles for SAM and suggests that links to specific methylation targets are essential to understand its biological importance in life span and stress responses.

## Materials and methods

### *C. elegans* strain and culture, RNAi and stress applications

The N2 strain was used for wild type and all other strains are described in S5 Table. Culture conditions are as previously described [84–86]. For metabolomics and heat shock experiments, adults were bleached onto RNAi plates and progeny were grown to young adults at 15°C before heat shock. Animals were collected in S-Basal, washed, and frozen at −80°C until processing. For all other experiments, animals were maintained at 20°C. For heat shock survival assays, animals were bleached onto RNAi plates and allowed to grow to young adults at 15°C. Approximately 25–30 animals were moved onto 35 mm plates in triplicate, resulting in 75–90 animals in each biological replicate. These plates were placed in a 37°C incubator for 2 hours, then moved to 20°C for the remainder of the assay. Starting the next day, each plate was checked for dead animals by looking for movement and, failing that, gentle prodding of the worm. Dead worms were removed and scored. Animals that died from internally hatched embryos (bagging) or died from desiccation were excluded from the final scoring. Three independent, non-blinded biological replicates were carried out and Kaplan–Meier curves were generated with GraphPad Prism v8.0.

### Relative targeted metabolite profiling

**Sample preparation.** Aqueous metabolites for targeted liquid chromatography–mass spectrometry (LC–MS) profiling of 80 *C. elegans* samples were extracted using previously described protein precipitation method [88,89]. Briefly, samples were homogenized in 200 μL purified deionized water at 4°C, and then 800 μL of cold methanol containing 124 μM 6C13-glucose and 25.9 μM 2C13-glutamate was added. Internal reference standards were added to the samples to

monitor sample preparation. Next, samples were vortexed, incubated for 30 min at −20°C, sonicated in an ice bath for 10 min, centrifuged for 15 min at 18,000g at 4°C, and then 600 µL of supernatant was collected from each sample (protein precipitate was used to do BCA assay for data normalization). Lastly, recovered supernatants were dried on a SpeedVac and reconstituted in 0.5 mL of LC-matching solvent containing 17.8 µM 2C13-tyrosine and 39.2 3C13-lactate, and internal reference standards were added to the reconstituting solvent to monitor LC–MS performance. Samples were transferred into LC vials and placed in a temperature-controlled autosampler for LC–MS analysis.

## LC–MS assay

Targeted LC–MS metabolite analysis was performed on a duplex-LC–MS system composed of two Shimadzu UPLC pumps, CTC Analytics PAL HTC-xt temperature-controlled auto-sampler and AB Sciex 6500+ Triple Quadrupole MS equipped with ESI ionization source [89]. UPLC pumps were connected to the autosampler in parallel and were able to perform two chromatography separations independently from each other. Each sample was injected twice on two identical analytical columns (Waters XBridge BEH Amide XP) performing separations in hydrophilic interaction liquid chromatography mode. While one column was performing separation for MS data acquisition in ESI+ ionization mode, the other column was equilibrated for sample injection, chromatography separation, and MS data acquisition in ESI- mode. Each chromatography separation was 18 min (total analysis time per sample was 36 min), and MS data acquisition was performed in multiple-reaction-monitoring mode. The LC–MS system was controlled using AB Sciex Analyst 1.6.3 software. The LC–MS assay targeted 361 metabolites and 4 spiked internal reference standards. Measured MS peaks were integrated using AB Sciex MultiQuant 3.0.3 software. Up to 201 metabolites and 4 spiked standards were measured across the study set, and over 90% of measured metabolites were measured across all the samples. In addition to the study samples, two sets of quality control (QC) samples were used to monitor the assay performance and data reproducibility. One QC [QC(I)] was a pooled human serum sample used to monitor system performance, and the other QC [QC(S)] consisted of pooled study samples, which were used to monitor data reproducibility. Each QC sample was injected once for every 10 study samples. The data were highly reproducible with a median coefficient of variation (CV) of 4.0%.

## Metabolite data analysis

The raw metabolomics data was first preprocessed to ensure analysis of only high-quality data [90]. Metabolites that were not captured in 75% or more of the samples were excluded from the analysis. Next, metabolites solely from bacteria sources were identified and removed by comparing the mean of pooled blanks with the mean of pooled samples. QC was performed on a pooled sample where the same metabolites were repeatedly measured. If the CV was > 20%, metabolites were excluded from downstream analysis [90]. This dataset was uploaded onto the Metaboanalyst v6.0 web platform [91]. Metaboanalyst was used to normalize our data before identifying changing metabolites with respect to each SAM synthase or after heat shock. Missing values were handled with the default setting (remove feature with >50% missing values and otherwise replace with 1/5 of the minimum positive value for each variable). Data was ultimately normalized by sum and log10 transformation after examining the distribution of each normalization combination [92]. This data was downloaded and fed into GraphPad Prism v10.0.0 where a two-way ANOVA was performed. False discovery rate was controlled with the Two-stage step-up method of Benjamini, Krieger, and Yekutieli and a desired false discovery rate of 0.05. Graphs were visualized in Prism except for the PCA was performed in R v4.2.1.

## Lipidomics

*C. elegans* samples were extracted using a modified Folch procedure including Internal Standards (Splash Mix; Avanti Polar Lipids). Samples were resuspended based on sample weight. *LC–MS/MS:* Thermo Accucore C30 150 mm analytical column was used in the positive and negative ionization modes to acquire data. Data were analyzed using LipidSearch 5.1. Background filtering was performed using solvent blanks (no extraction blanks submitted, which are the ideal sample

to filter out background). Data were filtered within the software according to pre-determined parameters within the software. Lipid Class data was used to generate a total lipid signal, then data shown were normalized to total lipid signal. Statistical analysis and visualization were performed in Graphpad prism.

### RNA sequencing

RNA for deep sequencing was purified by Qiagen RNAeasy. Duplicate samples were sent for library construction and sequencing at BGI (China) (S2 Table; Figs 5 and 6) or triplicate samples were sequenced at Biostate AI (USA) (S3 Table; S4 Fig). Raw sequencing reads were processed using a Nextflow pipeline (https://github.com/DanHUMassMed/RNA-Seq-Nextflow). The raw read pairs were first aligned to *C. elegans* reference genome with ws245 annotation. The RSEM method was used to quantify the expression levels of genes and DEseq was used to produce differentially expressed gene sets with more than a 2-fold difference in gene expression, with replicates being within 0.05 in a Students *T* test and a False Discovery Rate (FDR) under 0.01. Statistics were calculated with DeBrowser [93]. Venn Diagrams were constructed by BioVenn [94]. WormCat analysis was performed using the website https://www.wormcat.com/ [92] and the whole genome annotation version 2 (v2) and indicated gene sets. PCA was conducted by using *prcomp* in R and graphed with *ggplot* in R studio. The GEO accession number is GSE288260 for S2 Table, Figs 5 and 6 and is GSE308504 for S3 Table, S4 Fig

### Microscopy: TMRE staining

Plates with TMRE were prepared by dripping 10 μM TMRE onto the bacterial lawn and allowing the plate to dry in a hood immediately before use. Embryos from 20 to 25 animals were recovered by hypochlorite treatment on TMRE plates and imaged when the animals reached the young adult stage.

### Microscopy: confocal imaging

*C. elegans* were imaged on a Leica SPE6 or Nikon Spinning Disc as noted with identical gain or exposure settings within each experiment. Images were quantitated in FIJI with the following pipelines: (1) TMRE staining with MitoMapr [95], (2) PunctaProcess (https://github.com/DanHUMassMed/puncta_process_plugin) for counting GFP::LGG-1, and (3) Colocalization of GFP::LGG-1 and TOMM-20::mKate with the Fiji Image calculator, followed by Puncta Process to quantify the overlapping pixels. Archived code is located on Zenodo (https://doi.org/10.5281/zenodo.17495841). Graphs and statistical analysis were completed in Graphpad Prism.

### Microscopy: transmission electron microscopy

A suspension of *C. elegans* (tetramizole/ M9) were placed in the 100 μm deep side of type A 6 mm Cu/Au carriers (Leica), sandwiched with the flat side of type B 6 mm Cu/Au carriers (Leica), and frozen in a high-pressure freezer (EM ICE, Leica). This was followed by Freeze Substitution (FS) in a Leica EM AFS2 unit cooled down to −90°C. For FS Procedure and Embedding, the sandwiches with frozen samples were transferred under LN2 into cryovials containing frozen FS media (Acetone containing 1% OsO4/1% glutaraldehyde/1% water). The vials are placed into the precooled AFS2 unit, the lids are screwed loosely onto the vials to permit safe evaporation of excess N2 gas and after about 1 h the lids are tightened, and FS is started. During the FS, the temperatures increased to—90°C for 36–48 h to 60°C at a rate of 5°C per hour (6 h). After −60°C for 6 h, samples were warmed up to −30°C at a rate of 5°C per hour (6 h). Similarly, after 3 h at—30°C samples were warmed up to 0°C at a rate of 5°C per hour over 6 h. Finally, samples were warming up to 20°C at a rate of 5°C per hour. After reaching room temperature, the samples were rinsed three times with acetone (10 min each), one time with propylene oxide (PO) and infiltrated in a 1:1 mixture of PO:TAAB Epon overnight. The next day, specimens were transferred to 100% TAAB Epon, incubated 2 h at RT then transferred to fresh TAAB Epon in an embedding mold or embedded flat between sheets of Aclar, plastic, and moved to the oven to polymerize at 60°C for 48 h.

## Immunoblots

*C. elegans* (wild type and *sams-1(lof)*) were grown to adulthood, washed, and frozen at −80°C. Frozen pellets were lysed by sonication in RIPA buffer (50 mM Tris-HCl: at pH 7.4, 150 mM NaCl, 1% Triton X-100, 0.5% Sodium deoxycholate, 0.1% SDS (sodium dodecyl sulfate), 1 mM EDTA, 1 mM DTT, Complete Protease inhibitors). Extracts were centrifuged for 15 min at 12,000 rpm, and the supernatants were normalized for protein concentration. Samples were run on NuPAGE 4%–12% Bis-Tris gels using the MES buffer system and transferred to nitrocellulose membranes. Blots were blocked in 5% powdered milk and probed with primary antibodies against TOMM20 (ab78547), ETFB (17925–1-AP), and α-tubulin (AB-1157911). Secondary antibodies (w402b and ab97051) were used, and signals were detected using ECL (WBKLS0050) and visualized on an iBright 1,500 system.

## MitoIPs and oxygen consumption assays

Synchronized *ges-1*::TOMM-20::mKate::HA animals (50,000–70,000 worms at the late L4/young adult stage) were collected from control and *sams-1* RNAi plates. Animals were washed three times with S. Basal buffer and once with autoclaved Milli-Q water. The pellet was resuspended in 3 ml ice-cold Mitochondrial Isolation Buffer (MIB: 70 mM sucrose, 210 mM D-mannitol, 5 mM HEPES, 1 mM EGTA, 0.5% BSA; supplemented with protease inhibitor cocktail before use), then were homogenized on ice using a Teflon homogenizer (300 strokes). The lysate was centrifuged at 200$g$ for 5 min at 4°C, the supernatant was further clarified at 800$g$ for 10 min as in [35]. Finally, mitochondria were pelleted at 12,000$g$ for 10 min at 4°C. The mitochondrial pellet was gently resuspended in ice-cold mitochondrial assay solution (1× MAS: 70 mM sucrose, 220 mM D-mannitol, 10 mM $KH_2PO_4$, 5 mM $MgCl_2$, 2 mM HEPES, 1 mM EGTA, 0.2% BSA; supplemented with 2.5 mM succinate, 5 mM glutamate, 5 mM pyruvate, 2 mM ADP, and protease inhibitors). Protein concentration was determined by Bradford assay, and equal amounts of mitochondrial protein were used for immunoprecipitation with HA beads. Beads were pre-washed 3× with MAS and then incubated with mitochondrial suspensions for 90 min at 4°C, and washed once with MAS buffer. IP was verified by visualization of mKATE puncta on a Leica SP6 confocal. Beads were resuspended for Seahorse analysis in MAS supplemented with Glutamate (5 mM), Succinate (2.5 mM), Pyruvate (5 mM), and ADP (2 mM). All buffers and substrate stock solutions were adjusted to pH 7.2 with KOH

## Supporting information

**S1 Fig. Schematic of metabolic pathways represented in data from targeted metabolomics comparing SAM synthase knockdown in basal and heat-shocked *C. elegans*, based on WormPaths diagrams.** Pathways were adapted from WormPaths [17]. Colored boxes correspond to metabolites shown in individual graphs. Bolded metabolites are represented in targeted metabolomics.
(TIF)

**S2 Fig. Comparison of individual metabolite levels from targeted metabolomics.** Box and whisker plots showing individual metabolites from targeted metabolomics comparing heat-shocked *sams-1 and sams-4(RNAi)* animals **(A–L)**. Colored boxes show location of selected metabolites on S2 Fig. Significance was determined by two-way repeated measures ANOVA. ns: $q$-value ≥ 0.05 *: $q$-value < 0.05, **: $q$-value < 0.01, ***: $q$-value < .001, ****: $q$-value < 0.0001. Color blocks map to areas on metabolic map (Figs S2, 2 and 3). Underlying data is in S1 Table.
(TIF)

**S3 Fig. Distribution of *sams-1(lof)* upregulated genes during aging in GO Mitochondrion.** Heat map of GO: 00005739 (Mitochondrion) category. Underlying data is in S2 Table.
(TIF)

**S4 Fig. RNA sequencing of *sams-1(lof)* animals reveals downregulation of nuclear encoded mitochondrial proteins. (A)** PCA plot showing groupings of three replicates from WT and *sams-1(lof)* animals at D1 and D7 of adulthood. **(B)** Bubble chart showing WormCat category 1 and 2 enrichments compared in WT and *sams-1(lof)* animals at D1 and D7 of adulthood. FPKMs from mitochondrial genes **(C–F)**, *fat-7* **(G)**, a lipid metabolic gene that is induced when *sams-1* is reduced [66], mitoUPR and autophagy genes **(H–K)** and other stress-related transcription factors **(L–N)**. Error bars show standard deviation and the *p*-adjust value calculated by Deseq2 shows significance including a false discovery rate with * $p < 0.01$, ** $p < 0.005$, *** $p < 0.001$. Underlying data is in S3 Table.
(TIF)

**S5 Fig. Adult release of choline rescue has limited effects on mitochondrial stress or autophagy gene expression. (A)** Schematic showing experimental setup for generating populations of D1 and D7 animals for RNA preparation. qRT-PCR comparing *fat-7* **(B)**, *tomm-20* **(C)**, *Y111B2A.2/COX6C* **(D)**, and *mrpl-18* **(E)**. **(F)** Schematic showing experimental paradigm for choline drop experiment. qRT-PCR assays expression in animals maintained on choline, or where choline was removed as animals transitioned to adulthood comparing *fat-7* **(G)**, *hsp-6* **(H)**, *lgg-1* **(I)**, and *lgg-2* **(J)**. Significance calculated between pairs by the Student's test with Welch's correction and shown with * $p < 0.01$, ** $p < 0.005$, *** $p < 0.001$. Underlying data is in S1 Data.
(TIF)

**S6 Fig. Changes in mitochondrial morphology and lipid levels occur after loss of *sams-1*.** LCMS analysis comparing lipid class levels after *sams-1, sams-4* or *pcyt-1(RNAi)* for TG (triglycerides) **(A)**, PE (phosphatidylethanolamines) **(B)**, diglycerides (DG, **C**), phosphatidylglycerols (PG, **D**), Acylcarnatines (AcCa, **E**), cardiolipins (CL, **F**), Ubiquinone isoforms (CoQ, **G**), and sphingolipids (SPH, **H**). Underlying data is in S4 Table. **(I)** JalView visualization of a Clustal alignment of *C. elegans strt*-related proteins with human STAR7. **(J)** Diagram showing localization of mitochondrial targeting sequences and the STAR domain.
(TIF)

**S7 Fig. Quantification of mitochondrial changes after knockdown of PC synthesis or transport enzymes.** TMRE intensity comparison in wild-type and *sams-1(RNAi)* animals raised with and without dietary choline **(A)** or after *pmt-2* **(B)**, *pcyt-1* **(C)** RNAi. **(D)** Quantitation of mitochondrial networks from intestinally expressed *ges-1*::TOMMM-20::mKate in control or *strt-1(RNAi)* treated animals. Significance calculated by the Mann–Whitney test is shown with * $p < 0.01$, ** $p < 0.005$, *** $p < 0.001$. Underlying data is in S1 Data.
(TIF)

**S8 Fig. Changes in mitochondrial morphology and cellular localization occur after loss of *sams-1*.** Spinning disc confocal projections of TMRE staining comparing basal and heat-shocked animals **(A)** or heat-shocked wild-type and *sams-1(lof)* animals exposed to *fzo-1(RNAi)* **(B)**. Quantitation of junction points is in **(C)**. **(D)** TEM images of Control, *sams-1* or *pcyt-1* RNAi animals comparing mitochondrial morphology, localization and localization within intestinal cells. Scale bar is 500 nm. **(E)** Confocal projections of animals with intestinal expression of an autophagy flux reporter, GFP::LGG-1::mKate [68] **(E)**. Quantitation is in **(F)**. **(G)** Model. Significance calculated by the Mann–Whitney test is shown with * $p < 0.01$, ** $p < 0.005$, *** $p < 0.001$. Underlying data is in S1 Data.
(TIF)

**S9 Fig. Limited evidence for transcriptional regulation of autophagy genes in *sams-1* animals.** Column graphs of FPKMs from RNAseq for autogphagy–related genes (blue: **A–F**), other stress-related transcription factors (yellow: **G–I**). Whiskers encompass standard deviation and the *p*-adjust value calculated by Deseq2 shows significance including a false discovery rate with * $p < 0.01$, ** $p < 0.005$, *** $p < 0.001$. Underlying data is in S2 Table.
(TIF)

**S10 Fig. Browser tracks showing H3K4me3 peaks from Godbole and colleagues eLife (2023) of autophagy-related genes after *sams-1* and *sams-4(*RNAi) for *hlh-30* (A), *pha-4* (B), *sqst-1*(C), *lgg-1* (D), *lgg-2* (E), *atg-4.1* (F), and *sodh-1* (G).** Locations for primer sets used in Lim and colleagues 2023 are boxed.
(TIF)

**S1 Table. Targeted metabolomics of Control, *sams-1(RNAi)* and *sams-4(RNAi)* animals in basal and heat shock conditions. Tab 1.** Metabolites detected by LCMS in targeted analysis. Each metabolite is cross-referenced by HMB ID (Human Metabolome Database), common name, pubChem and KEGG database IDs, and WormPaths [17] identification. Classyfire [96], RefMet [97] and WormPaths classifications are included where available. **Tab 2.** LogFold changes for each metabolite. Up (peach) or down (light blue) changes were considered significant if over 2-fold with a *p* value <0.05. **Tab 3.** Raw values for metabolomcis. **Tab 4.** Principal Components.
(XLSX)

**S2 Table. RNAseq comparing gene expression during aging in wild-type and *sams-1(lof)* animals.** Tabs contain Deseq output of two replicates, comparisons to published mitoUPR and GO: Mitochondrion gene list and comparisons of WormCat enrichment for categories 1, 2, and 3. **List of Tabs: Tab 1:** N2_sams1_D1_all, **Tab 2:** N2_sams1_D7_all, **Tab 3:** D1_D7_comp, **Tab 4**: GO-0005739, **Tab 5**: WormCat1, **Tab 6**: WormCat2. **Tab 7**: WormCat3. **Abbreviations**: NC: No change. **Tab 8:** Principal components. **Abbreviations**: NF: Not found, NS: Not significant, NV: No value.
(XLSX)

**S3 Table. RNA seq comparing gene expression during aging in wild-type and *sams-1(lof)* animals.** Tabs contain Deseq output of three independent replicates, and comparisons of WormCat enrichment for categories 1, 2, and 3. **List of Tabs: Tab 1:** N2_sams1_D1_all, **Tab 2:** N2_sams1_D7_all, **Tab 3**: WormCat1, **Tab 4**: WormCat2, **Tab 5**: WormCat3, **Tab 6:** Principal Components. **Abbreviations**: NC: No change, NF: Not found, NS: Not significant. NV: No value.
(XLSX)

**S4 Table. Lipid classes in Control, *sams-1, sams-4*, and *pcyt-1* animals.** Lipid classes detected by LCMS and normalized to total lipid levels. Significance determined by paired Students *T* test. **Tab 1:** Values for lipid class measurements. **Tab 2:** Filters for LCMS. **Tabs 3–11:** *t* tests for lipid classes in Figs 7B or S6A–S6H.
(XLSX)

**S5 Table. List of strains used in this study.**
(XLSX)

**S1 Data. Raw data files for Figs 6D, 6F/7D, 6G, 6J, 6H/S8A, 7A, 7F, 7H, 8F, 8G, 8B-8E, S5G–S5J, and S7B.**
(XLSX)

**S1 Raw Image: Raw image file for Fig 6C.** For top row, blot was cut below the 55kd marker. In the middle row the blot was cut below the 55kd Marker and was the same gel as the TBA-1 probed blot. In the bottom row, the blot was cut above the 55kd marker. Although these are from the same gel, the blot was cut in the middle. Therefore, a break was placed in the figure. This is the same gel as the ETFB-1 probed blot. Imaging was done on an iQuant 1500 system blot.
(PDF)

## Acknowledgments

We would like to thank Drs. Cole Haynes, Marian Walhout, Eric B Baehrecke (UMASS Chan), and Miriam Greenberg (Wayne State) for helpful discussion. We appreciate the assistance of Dr. Marie Bao, Life Science Editors, for help in preparing the manuscript, along with Dr. Nils Grotehans and members of the Walker lab for helpful comments. We thank the

lab of Dr. Jin Zhang (UMASS) for use of the Nikon Spinning Disc for confocal microscopy and to the UMASS Metabolomics core for lipidomics (RRID:SCR_027036). Metabolomics were also performed at the University of Washington Nathan Shock Center of Excellence in the Basic Biology of Aging and transmission electron microscopy at the Harvard Medical School microscopy core. Some strains were provided by the CGC, which is funded by NIH Office of Research Infrastructure Programs (P40 OD010440).

## Author contributions

**Conceptualization:** Athena L. Munden, Dominique S. Lui, Adwait A. Godbole, Daniel E. L. Promislow, Dana L. Miller, Amy K. Walker.

**Data curation:** Athena L. Munden, Arjamand Mushtaq, Dominique S. Lui, Kasturi Biswas, Rachel M. Walker, Leah H. Crowley, Amy K. Walker.

**Formal analysis:** Athena L. Munden, Arjamand Mushtaq, Dominique S. Lui, Kasturi Biswas, Rachel M. Walker, Leah H. Crowley, Caroline A. Lewis, Danijel Djukovic, Amy K. Walker.

**Funding acquisition:** Jessica B. Spinelli, Daniel E. L. Promislow, Dana L. Miller, Amy K. Walker.

**Investigation:** Athena L. Munden, Arjamand Mushtaq, Dominique S. Lui, Katherine M. Edwards, Kasturi Biswas, Rachel M. Walker, Leah H. Crowley, Matthew J. Fanelli, Thien-Kim Nguyen, Maria Ericsson, Adwait A. Godbole, John A. Haley, Caroline A. Lewis, Jessica B. Spinelli, Benjamin Harrison, Daniel Raftery, Danijel Djukovic, Dana L. Miller, Amy K. Walker.

**Project administration:** Amy K. Walker.

**Resources:** Maria Ericsson, Caroline A. Lewis, Jessica B. Spinelli, Daniel Raftery, Danijel Djukovic, Daniel E. L. Promislow, Dana L. Miller.

**Software:** Daniel P. Higgins.

**Supervision:** Daniel E. L. Promislow, Amy K. Walker.

**Visualization:** Amy K. Walker.

**Writing – original draft:** Athena L. Munden, Dominique S. Lui, Katherine M. Edwards, Daniel P. Higgins, Matthew J. Fanelli, Thien-Kim Nguyen, Maria Ericsson, Adwait A. Godbole, John A. Haley, Caroline A. Lewis, Jessica B. Spinelli, Benjamin Harrison, Daniel Raftery, Danijel Djukovic, Daniel E. L. Promislow, Dana L. Miller, Amy K. Walker.

**Writing – review & editing:** Athena L. Munden, Dominique S. Lui, Katherine M. Edwards, Matthew J. Fanelli, Thien-Kim Nguyen, Maria Ericsson, Adwait A. Godbole, John A. Haley, Caroline A. Lewis, Jessica B. Spinelli, Benjamin Harrison, Daniel Raftery, Danijel Djukovic, Daniel E. L. Promislow, Dana L. Miller, Amy K. Walker.

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
