## [Editor Report · Decision Letter 0]

16 Feb 2025

Dear Dr Walker,

Thank you for submitting your manuscript entitled "Functional Specialization of S-Adenosylmethionine Synthases Links Phosphatidylcholine to Mitochondrial Function and Stress Survival" for consideration as a Research Article by PLOS Biology. Please accept my apologies for the delay in getting back to you with feedback as we consulted with an academic editor about your submission.

Your manuscript has now been evaluated by the PLOS Biology editorial staff, as well as by an academic editor with relevant expertise, and I am writing to let you know that we would like to send your submission out for external peer review.

Once your full submission is complete, your paper will undergo a series of checks in preparation for peer review. After your manuscript has passed the checks it will be sent out for review. To provide the metadata for your submission, please Login to Editorial Manager (https://www.editorialmanager.com/pbiology) within two working days, i.e. by Feb 18 2025 11:59PM.

Kind regards,

Richard

Richard Hodge, PhD

rhodge@plos.org

PLOS

---

## [Decision Letter · Decision Letter 1]

9 May 2025

Dear Dr Walker,

Thank you for your continued patience while your manuscript "Functional Specialization of S-Adenosylmethionine Synthases Links Phosphatidylcholine to Mitochondrial Function and Stress Survival" was peer-reviewed at PLOS Biology. Please accept my sincere apologies for the delays that you have experienced during the peer review process. Your manuscript has now been evaluated by the PLOS Biology editors, an Academic Editor with relevant expertise, and by two independent reviewers.

In light of the reviews, which you will find at the end of this email, we would like to invite you to revise the work to thoroughly address the reviewers' reports.

As you can see, both reviewers think the manuscript is interesting but raise overlapping concerns with the strength of the mechanistic data and lack of functional validation to support the findings. Specifically, Reviewer #1 asks for experiments to test whether lifespan extension is independent of autophagy and raises technical concerns with the mitochondrial imaging data. In addition, Reviewer #2 asks for the additional functional validation for the link between heat shock stress and SAM synthesis and requests several experiments that provide additional mechanistic insight for the model.

In addition, we noted that in their review, Reviewer #1 indicated that the paper could be restructured to a Resource Article. After discussions with the Academic Editor, we do not think that the manuscript would be a fit in this format at the journal and we would continue to consider a revised version as a conventional Research Article.

Given the extent of revision needed, we cannot make a decision about publication until we have seen the revised manuscript and your response to the reviewers' comments. Your revised manuscript is likely to be sent for further evaluation by all or a subset of the reviewers.

**IMPORTANT - SUBMITTING YOUR REVISION**

*Re-submission Checklist*

*Published Peer Review*

*PLOS Data Policy*

*Blot and Gel Data Policy*

Best regards,

Richard

Richard Hodge, PhD

rhodge@plos.org

REVIEWS:

Reviewer #1: In this manuscript, Munden et al studied the effect of sam synthase gene knockdowns on changes to metabolism and gene expression. The authors created a large dataset that is sure to be quite valuable to many different fields including metabolism and aging biology. The manuscript is also very well-written and easy to follow, with very nice graphics and schematics that can guide even those outside the field to follow the data. While the dataset and Figs. 1-6 are exceptional, one minor concern that I have is that there are many brazen conclusions made solely based on RNA-seq data without any experimental follow-up, which can be a bit dangerous to posit to the field as conclusions. The authors should either experimentally validate their claims or significantly tone down these conclusions, explicitly stating they are only speculations based on transcriptomics data, and not experimentally validated conclusions. This is especially true since the RNAseq seems to include only 2 biological replicates, which is pretty low for transcriptomics that is highly variable, especially in bulk C. elegans RNAseq. Finally, there are several major technical issues with the mitochondrial imaging that need to be corrected. Personally, I can't really see the value of the mitochondrial data in Fig. 7-8. The data has many technical issues and the conclusions cannot really be made based on this data. These issues are severely diminishing an otherwise impactful and extremely valuable dataset. I wonder if it would be better for this to be restructured to be a resource paper that provides the metabolomics and RNAseq data to the field and remove Fig. 7-8 that are full of technical difficulties that make conclusions very hard. I leave it up to the authors to decide whether the comments made below can be addressed to improve Fig. 7-8 or to remove them entirely and make this a data-based resource paper.

Major concerns:

-RNAseq data, especially C. elegans RNAseq data can be quite variable, as is very apparent in the PCA of Fig. 4B. It seems like the authors only conducted 2 replicates each of the RNAseq and with the exception of sams-1 RNAi at day 1, the other samples do not cluster very well. The clustering in Fig. 4C is definitely much better but I would highly recommend at least one more replicate is conducted to ensure the data is robust. This is especially important considering the many strong conclusive arguments the authors are making solely based on the RNAseq data without any experimental validation of the data (e.g., that hsp-6 levels are unchanged in the sams-1 RNAi condition, directly in opposition to previously published hsp-6p::GFP data, arguing that this is an artifact of the transgene reporter; arguing against a role for sams-1 RNAi in induction of autophagy, etc.).

-To make an argument that lifespan extension of sams-1lof is independent of autophagy based on RNAseq data is incorrect. While many autophagy-related genes (ATGs) are transcriptionally upregulated during prolonged autophagy, short-term or acute activation of autophagy can occur primarily through post-translational modifications and signaling cascades. Therefore, it is entirely possible that even with a lack of autophagy gene activation, there is significant autophagy activation and that the lifespan extension is dependent on autophagy. Either this conclusion needs to be removed or experimentally tested (e.g., disrupt autophagy and show that lifespan extension still occurs in sams-1lof).

-Because there is no strain list available, I am not sure which TOMM-20::mKate strain was being used. Was this an endogenously tagged TOMM-20 or an overexpression? If it is not endogenously tagged, the authors cannot use this as a proxy to measure mitochondrial mass since there is no way to distinguish between changes in transgene expression or mislocalization of an overexpressed mitochondrial protein. The authors either need to remove this or experimentally validate in an appropriate way (e.g., biochemically by probing for an endogenous mitochondrial-localized protein like porin, or quantifying mtDNA).

-If the authors are arguing a marked decrease in mitochondrial mass based on their TOMM-20::mKate, why is TMRE staining in Fig. 7C not lower? However, TMRE staining in Fig. 7F for sams-1 is actually lower, then it's higher in Fig. 7H? This is extremely confusing why the sams-1 RNAi vs. control is so dramatically different in each figure. Why does sams-4 RNAi have lower signal? Are all these differences because of decreased membrane potential or mitochondrial mass or both? These inconsistencies need to be addressed.

-The authors are arguing that sams-1lof animals do not have higher thermotolerance in fzo-1 depletion because they say sams-1lof already has fragmented mitochondria. If this is the case, then is drp-1 RNAi not reducing thermotolerance because drp-1 knockdown cannot result in increased fusion of mitochondria in the sams-1lof animal? Please perform the imaging of fzo-1/sams-1lof and drp-1/sams-1lof to ensure that the conclusions about linking mitochondrial fragmentation to thermotolerance are actually valid.

-Please include a strain list. This is essential to ensure that the experiments can be reproduced by future studies.

Minor concerns:

-Page 17 - where Fig. 7C, and D are referenced, RNAi should have a capital A.

-For referencing hsp-6p::GFP and hsp-4p::GFP please put the "p" to indicate promoter to prevent confusion with the HSP-6::GFP and HSP-4::GFP translational reporters.

Reviewer #2: The authors analyzed the effects of SAMS-1 and SAMS-4 on metabolites and found that SAMS-1 affects methionine metabolism, polyamine metabolism, and mitochondrial metabolites. Furthermore, analysis of gene expression in the sams-1 mutant indicated a reduction in nuclear-encoded mitochondrial gene expression and induction of mitophagy. The authors have found an intriguing phenomenon, but there also remain many issues that need to be resolved.

(1) Figs. 1E, 2B: Heat shock stress induced high accumulation of SAM in wild-type, sams-1, and sams-4 mutants. The reason for this is that heat shock stress activates SAM synthesis, but it is unclear whether SAM synthesis is actually promoted or not. The link between increased methylation and increased SAM synthesis is not clear, especially since the same group's 2023 paper published in eLife showed that sams-4 mutants exhibited no increase in H3K4Me3 levels upon heat shock stress. The authors should examine the flux analysis of SAM when heat shock stress is applied in wild strains, sams-1, and sams-4 mutants, and what happens to the accumulation of SAM when heat shock stress is administered in the sams-1 mutant under knockdown of sams-4.

(2) Figs. 1, 2: Did spermidine levels change in metabolome analysis? It has been shown that spermidine contributed to lifespan extension in Caenorhabditis elegans (Nat Cell Biol. 2009 doi: 10.1038/ncb1975. Epub 2009 Oct 4. Tobias Eisenberg et al.). Therefore, does impairing spermidine synthesis affect the phenotype of sams-1 mutant strains? It has been reported that a decreased methionine level promoted choline synthesis and decreased SAM synthesis, which in turn enhanced putrescine and spermidine synthesis (Biomolecules. 2023 doi: 10.3390/biom13030471. Sayaka Harada et al.). Could the same thing be happening in the sams-1 mutant strain?

(3) Figs. S2B, S2C: Why do sams-1 mutants accumulated high levels of polyamines despite decreased accumulation of SAM?

(4) What is the mechanism by which abnormal PC synthesis affects MT function? Have the authors investigated the possibility that MT function is affected via lipid exchange by the ER-mitochondria connection?

(5) What is the mechanism by which SAMs synthesized by SAMS-1 and SAMS-4 are primarily used for metabolism and methylation, respectively? Is it possible that the expression of SAMS-4 in the gonads affects methylation? Experiments with L1, which is unaffected by the gonads, may help clarify this.

---

## [Decision Letter · Decision Letter 2]

20 Oct 2025

Dear Dr Walker,

Thank you for your patience while we considered your revised manuscript "Functional Specialization of S-Adenosylmethionine Synthases Links Phosphatidylcholine to Mitochondrial Function and Stress Survival" for publication as a Research Article at PLOS Biology. This revised version of your manuscript has been evaluated by the PLOS Biology editors, the Academic Editor and the original reviewers.

Based on the reviews, I am pleased to say that we are likely to accept this manuscript for publication, provided you satisfactorily address the following data and other policy-related requests that I have provided below (A-G):

(A) We routinely suggest changes to titles to ensure maximum accessibility for a broad, non-specialist readership. In this case, we would suggest a minor edit to the title, as follows. Please ensure you change both the manuscript file and the online submission system, as they need to match for final acceptance:

“Distinct S-adenosylmethionine synthases link phosphatidylcholine to mitochondrial function and stress survival"

(B) You may be aware of the PLOS Data Policy, which requires that all data be made available without restriction: http://journals.plos.org/plosbiology/s/data-availability. For more information, please also see this editorial: http://dx.doi.org/10.1371/journal.pbio.1001797

-Supplementary files (e.g., excel). Please ensure that all data files are uploaded as 'Supporting Information' and are invariably referred to (in the manuscript, figure legends, and the Description field when uploading your files) using the following format verbatim: S1 Data, S2 Data, etc. Multiple panels of a single or even several figures can be included as multiple sheets in one excel file that is saved using exactly the following convention: S1_Data.xlsx (using an underscore).

-Deposition in a publicly available repository. Please also provide the accession code or a reviewer link so that we may view your data before publication.

Figure 1B-E, 2B-E, 3B-E, 4B-E, 4H-K, 5C-F, 5H-I, 6D, 6F-H, 6J, 7B, 7D, 7F, 7H, 8A-B, 8F-G, S2A-L, S4A, S4C-N, S5B-E, S5G-J, S6A-H, S7A-D, S8C, S8F, S9A-I

(C) Please also ensure that each of the relevant figure legends in your manuscript include information on *WHERE THE UNDERLYING DATA CAN BE FOUND*, and ensure your supplemental data file/s has a legend.

(D) We require the original, uncropped and minimally adjusted images supporting all blot and gel results reported in the following Figures:

Figure 6C

We will require these files before a manuscript can be accepted so please prepare and upload them now. Please carefully read our guidelines for how to prepare and upload this data: https://journals.plos.org/plosbiology/s/figures#loc-blot-and-gel-reporting-requirements. *We note that Figure S1 looks to be provide the raw blots for Figure 1E, but we ask that the raw and uncropped images are provided in a separate raw image file in the Supplementary Information.

(E) Please ensure that your Data Statement in the submission system accurately describes where your data can be found and is in final format, as it will be published as written there (i.e. GSE288260 is now publicly available).

(F) Per journal policy, if you have generated any custom code during the course of this investigation, please make it available without restrictions. Please ensure that the code is sufficiently well documented and reusable, and that your Data Statement in the Editorial Manager submission system accurately describes where your code can be found.

(G) Please ensure that you are using best practice for statistical reporting and data presentation. These are our guidelines https://journals.plos.org/plosbiology/s/best-practices-in-research-reporting#loc-statistical-reporting and a useful resource on data presentation https://journals.plos.org/plosbiology/article?id=10.1371/journal.pbio.1002128

- If you are reporting experiments where n ≤ 5, please plot each individual data point.

We expect to receive your revised manuscript within two weeks.

*Published Peer Review History*

*Press*

Best regards,

Richard

Richard Hodge, PhD

rhodge@plos.org

Reviewer remarks:

Reviewer #1 (Ryo Higuchi-Sanabria, identifies himself): The authors have put incredible effort into addressing all the reviewer comments, including massively increasing the sample size for the transcriptomics datasets, as well as furthering their experimental validation. I applaud the authors for such a thorough revision and believe this manuscript in its current form would be an incredible resource for the field.

Reviewer #2: The authors have adequately addressed my questions. Consequently, I consider this manuscript suitable for publication in PLOS Biology.

---

## [Editor Report · Decision Letter 3]

11 Nov 2025

Dear Amy,

On behalf of my colleagues and the Academic Editor, William Mair, I am pleased to say that we can accept your manuscript for publication, provided you address any remaining formatting and reporting issues. These will be detailed in an email you should receive within 2-3 business days from our colleagues in the journal operations team; no action is required from you until then. Please note that we will not be able to formally accept your manuscript and schedule it for publication until you have completed any requested changes.

PRESS

Best wishes, 

Richard

Richard Hodge, PhD

rhodge@plos.org

PLOS
